# Auto-inhibition of Mif2/CENP-C ensures centromere-dependent kinetochore assembly in budding yeast

Kerstin Killinger[1] (iD), Miriam Böhm[1] (iD), Philine Steinbach[1,†] (iD), Götz Hagemann[2] (iD), Mike Blüggel[3] (iD), Karolin Jänen[1], Simone Hohoff[1], Peter Bayer[3], Franz Herzog[2] & Stefan Westermann[1,*] (iD)

## Abstract

Kinetochores are chromatin-bound multi-protein complexes that allow high-fidelity chromosome segregation during mitosis and meiosis. Kinetochore assembly is exclusively initiated at chromatin containing Cse4/CENP-A nucleosomes. The molecular mechanisms ensuring that subcomplexes assemble efficiently into kinetochores only at centromeres, but not anywhere else, are incompletely understood. Here, we combine biochemical and genetic experiments to demonstrate that auto-inhibition of the conserved kinetochore subunit Mif2/CENP-C contributes to preventing unscheduled kinetochore assembly in budding yeast cells. We show that wild-type Mif2 is attenuated in its ability to bind a key downstream component in the assembly pathway, the Mtw1 complex, and that addition of Cse4 nucleosomes overcomes this inhibition. By exchanging the N-terminus of Mif2 with its functional counterpart from Ame1/CENP-U, we have created a Mif2 mutant which bypasses the Cse4 requirement for Mtw1 binding in vitro, thereby shortcutting kinetochore assembly. Expression of this Mif2 mutant in cells leads to mis-localization of the Mtw1 complex and causes pronounced chromosome segregation defects. We propose that auto-inhibition of Mif2/CENP-C constitutes a key concept underlying the molecular logic of kinetochore assembly.

**Keywords** Ame1; CENP-U; CCAN; nucleosome; spindle
**Subject Categories** Cell Cycle; Structural Biology
**The EMBO Journal (2020) 39: e102938**

## Introduction

To properly distribute their genomes during cell division, eukaryotic cells have to assemble kinetochores from hundreds of individual proteins. Kinetochore assembly needs to occur in a timely manner such that sister chromatid bi-orientation, a process which involves a stochastic error correction process for each sister kinetochore pair, can be initiated as soon as possible (Foley & Kapoor, 2013). On the other hand, an uncontrolled assembly of kinetochores, either at non-centromeric loci or entirely away from chromatin, would have deleterious consequences for the cell. Kinetochore assembly in different organisms follows a general hierarchical scheme (Fig 1): A CENP-A deposition machinery mediates the placement or replenishment of CENP-A nucleosomes specifically at centromeres, either by interpreting epigenetic signals or by coupling sequence-specific information to the CENP-A loading machinery (Catania & Allshire, 2014; Fukagawa & Earnshaw, 2014; McKinley & Cheeseman, 2016). CENP-A is known to specifically recruit several proteins of the inner kinetochore, now usually referred to as the constitutive centromere-associated network (CCAN) (Foltz et al, 2006; Okada et al, 2006; Hori et al, 2008). The CCAN in turn is responsible for recruitment of the microtubule binding outer kinetochore, including the KMN network (Cheeseman et al, 2006; DeLuca et al, 2006; Ciferri et al, 2008) (Fig 1). The KMN network is recruited to the kinetochore through two axes: (i) recruitment of Mis12c/Mtw1c to CENP-C in human or Mif2 and Ame1 in S. cerevisiae (Przewloka et al, 2011; Screpanti et al, 2011; Hornung et al, 2014), which in turn leads to recruitment of Ndc80c, or (ii) direct recruitment of Ndc80c via CENP-T/Cnn1 in human and yeast cells (Gascoigne et al, 2011; Schleiffer et al, 2012; Nishino et al, 2013). Two CCAN subunits, CENP-C and CENP-N, specifically recognize structural features that distinguish CENP-A- from H3 nucleosomes and contribute to downstream kinetochore assembly at centromeres (Carroll et al, 2009, 2010; Kato et al, 2013; Xiao et al, 2017). Among the CCAN subunits, CENP-C plays an especially important and highly conserved role: In most organisms investigated so far, CENP-C loss-of-function mutants cause severe defects in kinetochore assembly, with all kinetochore subunits with the exception of CENP-A failing to localize to centromeres (Carroll et al, 2010; Klare et al, 2015; Weir et al, 2016). Along the CENP-C polypeptide chain, an Mtw1c binding domain at the extreme N-terminus (Przewloka et al, 2011; Screpanti et al,

1 Department of Molecular Genetics, Faculty of Biology, Center of Medical Biotechnology, University of Duisburg-Essen, Essen, Germany
2 Department of Biochemistry, Gene Center, Ludwig-Maximilians-Universität München, München, Germany
3 Structural and Medicinal Biochemistry, Faculty of Biology, Center of Medical Biotechnology, University of Duisburg-Essen, Essen, Germany
*Corresponding author. Tel: +49 2011832733; E-mail: stefan.westermann@uni-due.de
†Present address: Institute of Medical Microbiology, University of Duisburg-Essen, Essen, Germany

2011), separate binding motifs for the CCAN subunits CENP-HIK (Klare *et al*, 2015), CENP-NL (Pentakota *et al*, 2017), and the "CENP-C signature motif" that interacts with the CENP-A-specific carboxy-terminus (Carroll *et al*, 2010; Kato *et al*, 2013) are arranged in a linear manner. Apart from the C-terminal cupin fold that mediates homo-dimerization of the molecule, CENP-C is predicted to be largely disordered in isolation (Cohen *et al*, 2008). With its ability to connect the KMN network, CCAN subunits, and the CENP-A nucleosome, CENP-C plays a central role in the molecular architecture of kinetochores. In addition, CENP-C is involved in the loading of new CENP-A and influences structure and properties of the CENP-A nucleosome (Guse *et al*, 2011; Fukagawa & Earnshaw, 2014; Falk *et al*, 2015). Whereas in humans, several CENP-A nucleosomes are present at every centromeric region (Bodor *et al*, 2014), centromeres of budding yeast only contain a single Cse4 nucleosome on which a single kinetochore is built (Furuyama & Biggins, 2007). Despite this difference between yeast kinetochores and those of higher eukaryotes, most kinetochore subunits are conserved from yeast to human, indicating similarities in their structure and function (Westermann & Schleiffer, 2013).

In budding yeast, Mif2/CENP-C is one of only three essential CCAN components. Interestingly, its essentiality is not based on its interaction with Mtw1c, as the Mif2 N-terminal motif is dispensable and its loss only causes benomyl hypersensitivity in otherwise wild-type yeast cells (Hornung *et al*, 2014). A recent study has demonstrated that Mtw1c binding by CENP-C is also dispensable in chicken DT40 cells (Hara *et al*, 2018). We have previously shown that the CENP-U/Q homologs Ame1 and Okp1 form a heterodimer and that Ame1, similar to Mif2, contains an N-terminal Mtw1c binding motif, which is essential for viability in budding yeast (Hornung *et al*, 2014; Schmitzberger *et al*, 2017) (Fig 1). As Mtw1c is the assembly hub for the outer kinetochore (Cheeseman *et al*, 2006; Maskell *et al*, 2010; Petrovic *et al*, 2010; Hornung *et al*, 2011), defining its recruitment through its two receptors Mif2 and Ame1 is highly important for understanding overall kinetochore assembly. Similar to Mif2, Ame1 has recently been shown to bind CENP-A nucleosomes, interacting with the Cse4-specific N-terminal domain (Anedchenko *et al*, 2019; Fischbock-Halwachs *et al*, 2019). How the two Mtw1c receptors co-ordinate the assembly of a higher-order

budding yeast kinetochore architecture, which contains multiple Mtw1c molecules, is not well understood.

Here, we show that the interaction between Mif2 with Mtw1c is controlled through an auto-inhibitory mechanism that prevents inappropriate binding. This inhibitory conformation of Mif2 is released upon its binding to Cse4 nucleosomes. In this manner, the Mif2 molecule not only functions as a major factor in kinetochore assembly, but also safeguards against inappropriate kinetochore assembly away from centromeric chromatin. Furthermore, we provide insights into the assembly of a higher-order kinetochore architecture through the elaboration of Mtw1c binding to Ame1 and Mif2, which our data indicate to be mutually exclusive. Instead of cooperating in the binding of the same Mtw1c molecule, Ame1 and Mif2 bind different molecules using a conserved binding motif, as the Mif2 N-terminus grafted onto Ame1 fully restores Ame1 function in cells.

## Results

### The N-termini of Ame1 and Mif2 bind competitively to the Mtw1 complex

Two different configurations of Mtw1c molecules bound to Ame1 and Mif2 receptors can be considered (Fig 1): Either Mif2 and Ame1 cooperate in the binding of a single Mtw1c molecule using distinct sites on head 1 that allow simultaneous binding (scenario 1), or they use an identical or largely overlapping binding site on head 1 and interact with separate Mtw1c molecules (scenario 2). To distinguish between these possibilities, we tested a potential competition between Ame1 and Mif2 in their binding to Mtw1c in a solid-phase binding assay in which GST or GST-Mif2-N$^{1-41}$ was incubated with the Mtw1-Nnf1 heterodimer (MN) in the presence of increasing concentrations of soluble Ame1-N peptide (aa 1–20). We decided to use the MN, because it has been shown that Dsn1 contains an auto-inhibitory loop that prevents Mtw1c from binding to Mif2 (Dimitrova *et al*, 2016). In agreement with scenario 2, Ame1-N was able to inhibit binding of MN to GST-Mif2-N in a dose-dependent manner (Fig 2A). To verify the competitive binding of Ame1 and Mif2 to

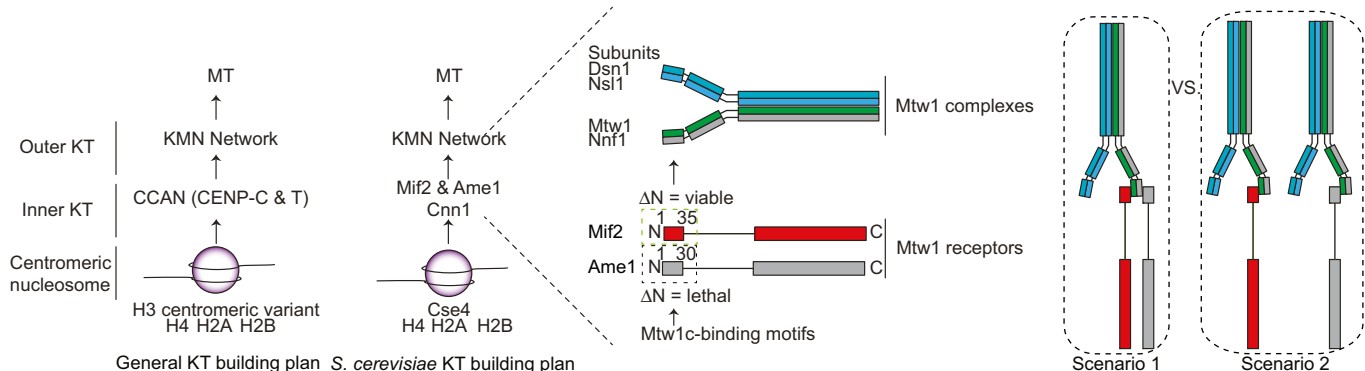

**Figure 1. Ame1 and Mif2 are Mtw1c receptors that promote outer kinetochore assembly.**

Left panel: Comparison of kinetochore assembly plans. Generally, inner kinetochores are represented by the constituted centromere-associated network (CCAN) of which Ame1 and Mif2 are the only essential subunits in *S. cerevisiae*. Right panel: Among other functions, Ame1 and Mif2 both act as receptors for the outer kinetochore component Mtw1c using short binding motifs at their N-termini. An Ame1ΔN is lethal, while a Mif2ΔN mutant is viable. Two different scenarios for Mtw1c-receptor configurations are indicated.

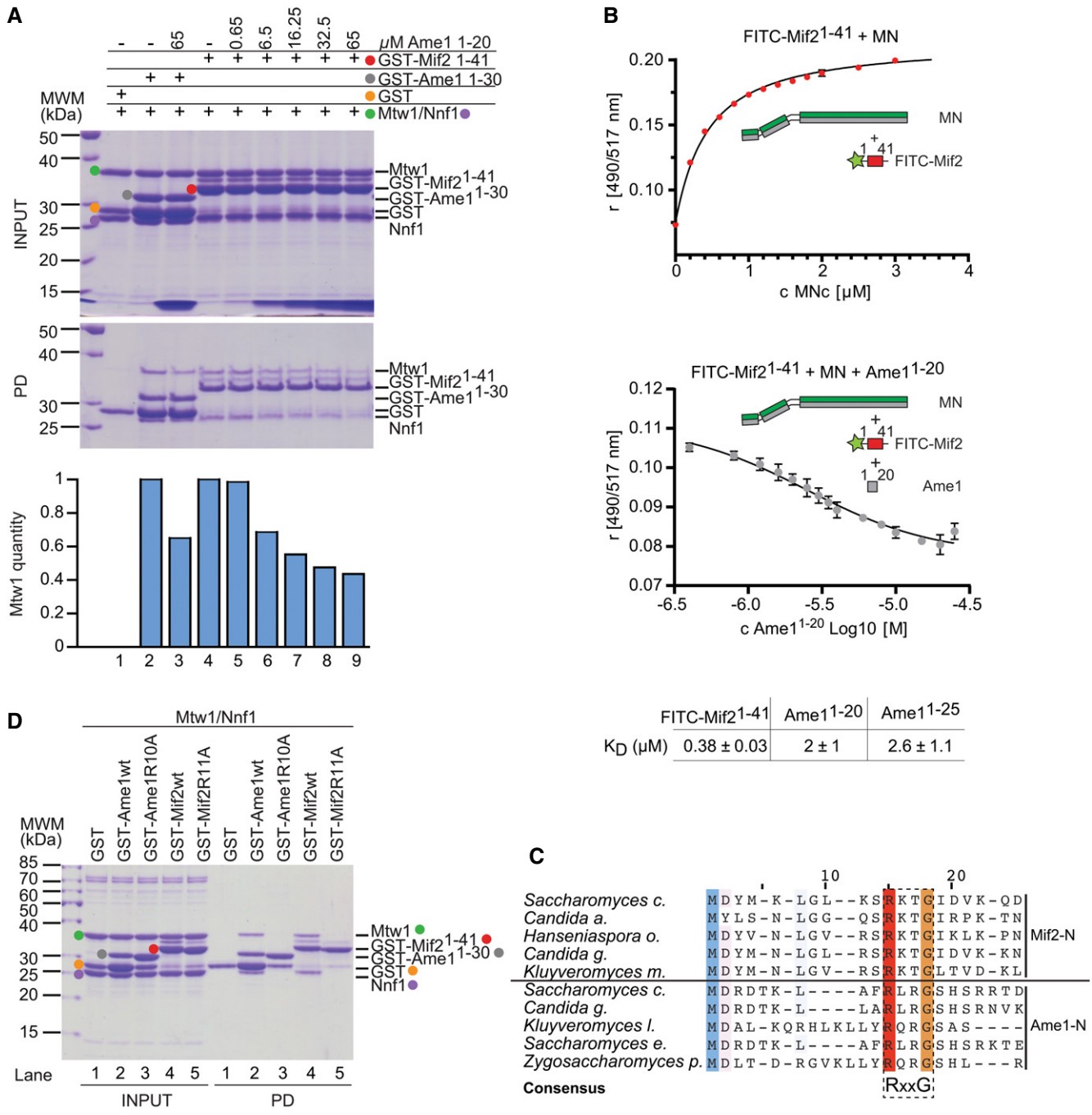

**Figure 2. Ame1 and Mif2 share overlapping Mtw1c binding motifs.**

A GST-Mif2$^{1-41}$ was immobilized on GSH Sepharose beads and incubated with Mtw1-Nnf1 complex (MN) in the presence of increasing concentrations of Ame1$^{1-20}$ peptide (lanes 4–9). GST-Ame1$^{1-30}$ served as control (lane 2 and 3). After washing, bound complexes were eluted by adding SDS sample buffer. Protein bands in Coomassie-stained gel are highlighted in the indicated color. Integrated density of Mtw1 bands was quantified using ImageJ and correlated to GST-Mif2 amounts in the same lane.

B Binding of FITC-Mif2$^{1-41}$ to MN was directly determined in a fluorescence polarization (FP) assay. This interaction at EC50 was than competed with increasing concentration of Ame1$^{1-20}$ unlabeled peptide. Both experiments were performed in triplicates of which the mean and standard deviation are shown.

C Multiple sequence alignment of the N-termini of Mif2 and Ame1 from distantly related yeast species. Conserved residues are highlighted. A dashed box indicates a conserved consensus sequence of RxxG.

D Equivalent point mutations replacing Arg11 in Mif2 or Arg10 in Ame1 with alanine prevent binding to the MN complex. Mif2 and Ame1 wild-type N-termini serve as a control, and protein bands are highlighted in the indicated color.

Mtw1c in solution, an N-terminal Mif2 peptide was fluorescently labeled with FITC and binding affinity determined directly in a fluorescence polarization (FP) assay. The FP assay yielded robust binding of Mif2-N to MN with a dissociation constant of $K_d = 0.38$ μM. FITC-Mif2 and MN were then pre-incubated at their previously determined half-maximal effective concentration (EC50), and the binding was competed with an unlabeled Ame1 peptide. A sigmoidal decrease in the Mif2-N FP signal indicated effective competition by Ame1-N. Two different peptides yielded similar dissociation constants of around $K_d = 2$ μM for the binding of Ame1-N to MN. We conclude that the N-termini of Mif2 and Ame1 bind competitively to the Mtw1c, and thus, their binding to a single Mtw1c must be mutually exclusive (Fig 2B).

A multiple sequence alignment of N-terminal sequences of Mif2 and Ame1 proteins revealed shared features. In particular, a stretch of four amino acids containing an invariant arginine and a glycine residue is conserved between different Mif2 and Ame1 N-termini. Notably, Arg11 of budding yeast Mif2 aligns well with Arg10 of Ame1, a residue whose mutation to aspartate was shown to be lethal (Hornung *et al*, 2014). This arginine residue is followed by a conserved glycine at position +3, leading to a consensus sequence of RxxG (Fig 2C). We predicted that mutating this arginine residue in either Mif2-N or Ame1-N should abolish binding to the MN complex. Indeed, we found that these point mutations led to a complete loss of binding under conditions in which the wild-type proteins clearly interact with the MN heterodimer (Fig 2D).

### Mutation of the Mif2 binding interface on Mtw1 is lethal and disrupts the interaction with AO

To probe Mtw1c interactions with its receptors *in vivo*, we used the recently published crystal structure of Mif2-N bound to head I of the Mtw1-Nnf1 heterodimer (Dimitrova *et al*, 2016) as a reference to design point mutations of residues implicated directly in Mif2-N binding (interface I mutants), residues previously predicted to affect Ame1-N binding (interface II mutants), or a combination of both (interface I+II mutants) (Fig 3A). We first tested the impact of these point mutations in cells. To this end, we introduced either wild-type or mutant Mtw1 alleles into a yeast strain in which the endogenous Mtw1 gene was fused to an FRB-tag, such that it can be anchored away from the nucleus upon addition of rapamycin. A clear prediction from scenario 2 is that interface I mutants should not be compatible with viability, even though the published crystal structure only implicates them in the binding of the non-essential Mif2 N-terminus (Hornung *et al*, 2014). We rather predict that in scenario 2, mutation of these residues will also abolish Ame1 binding and therefore be lethal. In addition, interface II mutations, which are predicted to affect the essential connection between Ame1 and Mtw1c, should decrease viability, as should the combined interface I+II mutants.

In full agreement with scenario 2, we found that all three Mtw1 alleles were unable to confer cell growth in the presence of rapamycin (Fig 3B). The inability of these alleles to support growth was not due to lack of protein expression, although the steady-state level was reduced compared to the wild-type as shown by Western blotting (Fig 3D). We additionally confirmed that Mtw1 interface I mutants indeed confer a lethal phenotype by introducing the respective mutations into a hemizygous Mtw1 deletion strain and

performing tetrad dissection. While haploid Mtw1-5A and 8A mutants were inviable, we could recover viable, albeit slow-growing, Mtw1-3A cells (Fig 3C). This somewhat milder phenotype of Mtw1-3A might be due to the fact that in this mutant, interface I is still intact, which will probably allow some residual binding of Mtw1c to Ame1. The more severe phenotype of Mtw1-3A in the FRB system compared to tetrad dissection might be due to the additional mutations in the complex genetic background of the anchor-away strain, which might make it more sensitive to the Mtw1-3A mutation.

As described above, the most straightforward explanation for the lethality of Mtw1 interface I mutants would be their interference with both Mif2-N and Ame1-N binding, thereby destroying the essential connection to Ame1. We verified this hypothesis using recombinantly expressed proteins in size-exclusion chromatography (SEC)-based binding assays *in vitro*. As shown before, bacterially expressed Ame1-Okp1 (AO) complex interacted efficiently with wild-type Mtw1c, leading to co-elution in the form of a broad peak. While recombinant Mtw1 containing the interface I 5A mutation was able to assemble into heterotetrameric Mtw1c, this mutant complex was no longer able to interact efficiently with AO during SEC (Fig 3E). As expected, mutation of interface II or a combination of interface I+II also abolished interaction with AO in a SEC experiment (Fig EV1). We conclude that mutation of residues, defined by the crystal structure to be involved in Mif2-N binding, also prevents Ame1-N binding. This result confirms that rather than occupying spatially separated binding sites on head I of a single Mtw1c molecule (scenario 1), Mif2-N and Ame1-N occupy identical or largely overlapping binding sites on separate Mtw1c molecules (scenario 2).

### Grafting the Mif2 N-terminus onto Ame1 restores yeast viability

Our previous experiments suggest that the N-termini of Mif2 and Ame1 are in principle functionally equivalent regarding Mtw1 binding *in vitro*. To test this idea further, we asked whether the Mif2 N-terminus can functionally replace the Ame1 N-terminus in yeast cells, or *vice versa*. This can be addressed experimentally by exchanging the extreme N-termini of both proteins (swap mutants, Fig 4A). To this end, we expressed Ame1 wild-type, Ame1-ΔN, or a swap mutant in which the amino-terminal 15 residues of Ame1 were replaced by Mif2-N$^{1-35}$ (Ame1$^{Mif2-N}$ swap) in a strain in which endogenous Ame1 was FRB-tagged. As described before, expression of Ame1-ΔN was unable to sustain viability in the presence of rapamycin. By contrast, the Ame1$^{Mif2-N}$ swap allele conferred robust growth indistinguishable from wild-type Ame1 (Fig 4B). Observed growth phenotypes could not be explained by different protein expression levels, as checked by Western blotting (Fig 4C). We confirmed this result outside of the genetic context of the anchor-away system, using gene replacement in a diploid strain followed by tetrad dissection. While the Ame1-ΔN mutant was inviable, Ame1$^{Mif2-N}$ swap yielded four viable spores, and the mutant cells grew indistinguishably from wild-type cells under all conditions tested (Fig 4D). The endogenous Mif2 N-terminus remained nonessential in these cells, as we were able to construct a viable Ame1$^{Mif2-N}$ Mif2-ΔN double mutant strain (data not shown).

To determine the cellular basis for the observed rescue, we marked kinetochores with Mtw1-GFP and spindle pole bodies with

Spc42-RedStar in the Ame1-FRB strains. Four hours after addition of rapamycin, cells lacking any rescue allele were characterized by weak Mtw1 signals or displayed bi-lobed signals that had not progressed into anaphase in large-budded cells. The Ame1-ΔN rescue allele yielded a milder phenotype characterized by an accumulation of large-budded cells with weakened Mtw1 signals in a metaphase configuration (accounting for about 70% of all cells).

Both Ame1 wild-type and Ame1$^{Mif2-N}$ swap rescue alleles completely suppressed these defects and the cells displayed normal level and localization of Mtw1-GFP, with unperturbed cell cycle progression (Fig 4E and F).

We conclude that there is no uniquely essential function of the Ame1 N-terminus, and either Ame1-N or Mif2-N, when connected to the rest of the Ame1 polypeptide, will suffice for cell viability.

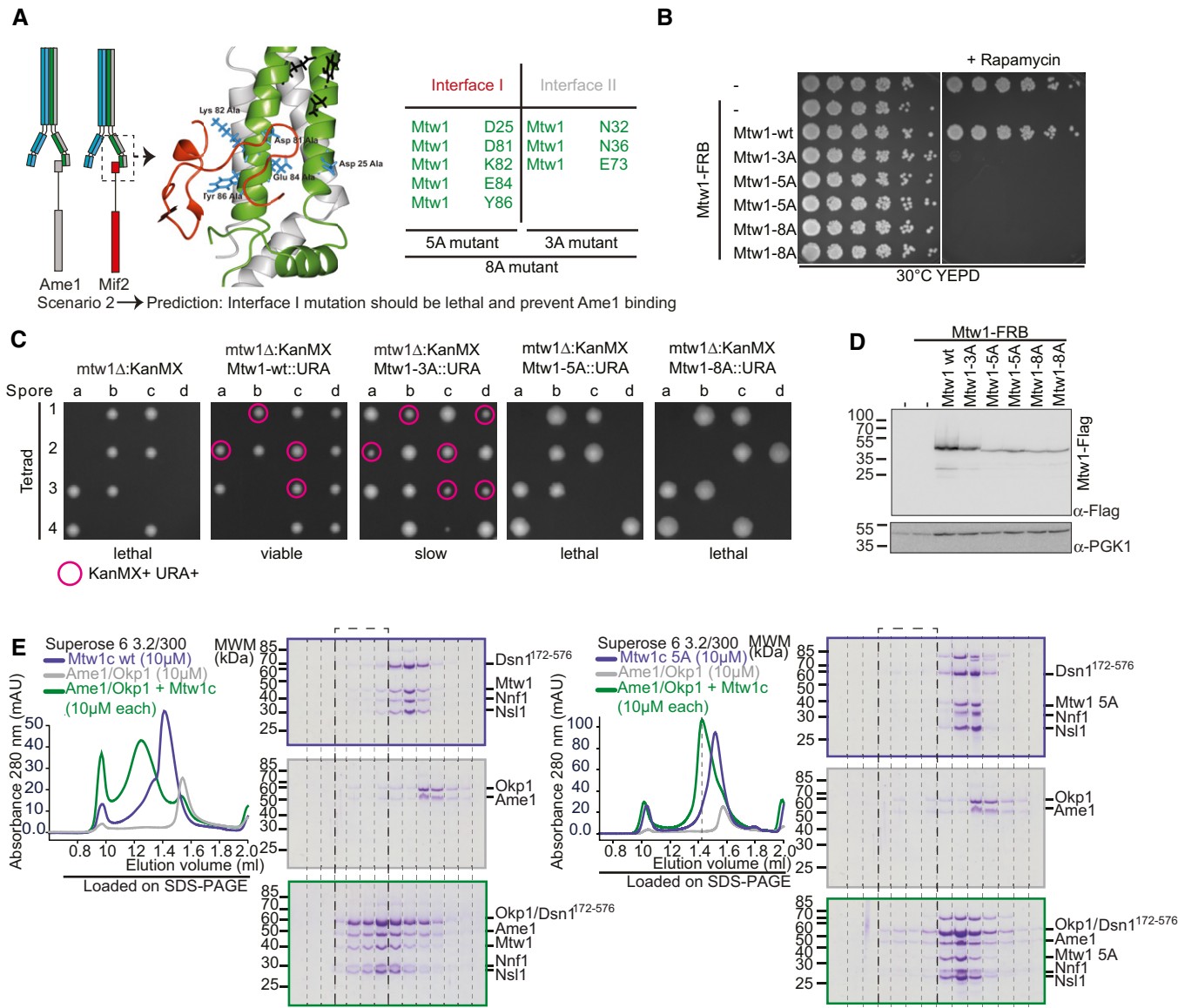

**Figure 3. Analysis of Mtw1 interface mutants *in vivo*.**

A  Structure of the Mtw1c head I bound to Mif2 and list of key residues constituting interface I (Mif2 binding) and interface II (putative Ame1 binding) from Dimitrova *et al*.

B  Analysis of these Mtw1 mutants *in vivo* using the anchor-away technique. Mtw1-FRB strains containing RPL13-FKBP12 for cytoplasmic anchoring and additionally harboring the indicated rescue alleles were plated in serial dilution on YEPD or YEPD + rapamycin plates incubated at 30°C.

C  Tetrad analysis of Mtw1-wt or mutants. Surviving spores in which the Mtw1-wt or mutant rescue construct was the only source of Mtw1 are indicated in pink circles.

D  Western blotting confirming the expression of Mtw1-6x-Flag wt or mutant proteins in the Mtw1-FRB strain background.

E  Analytical SEC runs and accompanying SDS–PAGE of Mtw1c-wt or Mtw1c-5A mutant (blue), AO (gray), or a stoichiometric combination of both complexes (green) at 10 μM each. Boxes indicate corresponding fractions in left and right panels. Note that the same Ame1-Okp1 elution profile and SDS–PAGE (middle panel) are displayed in both sets to improve clarity.

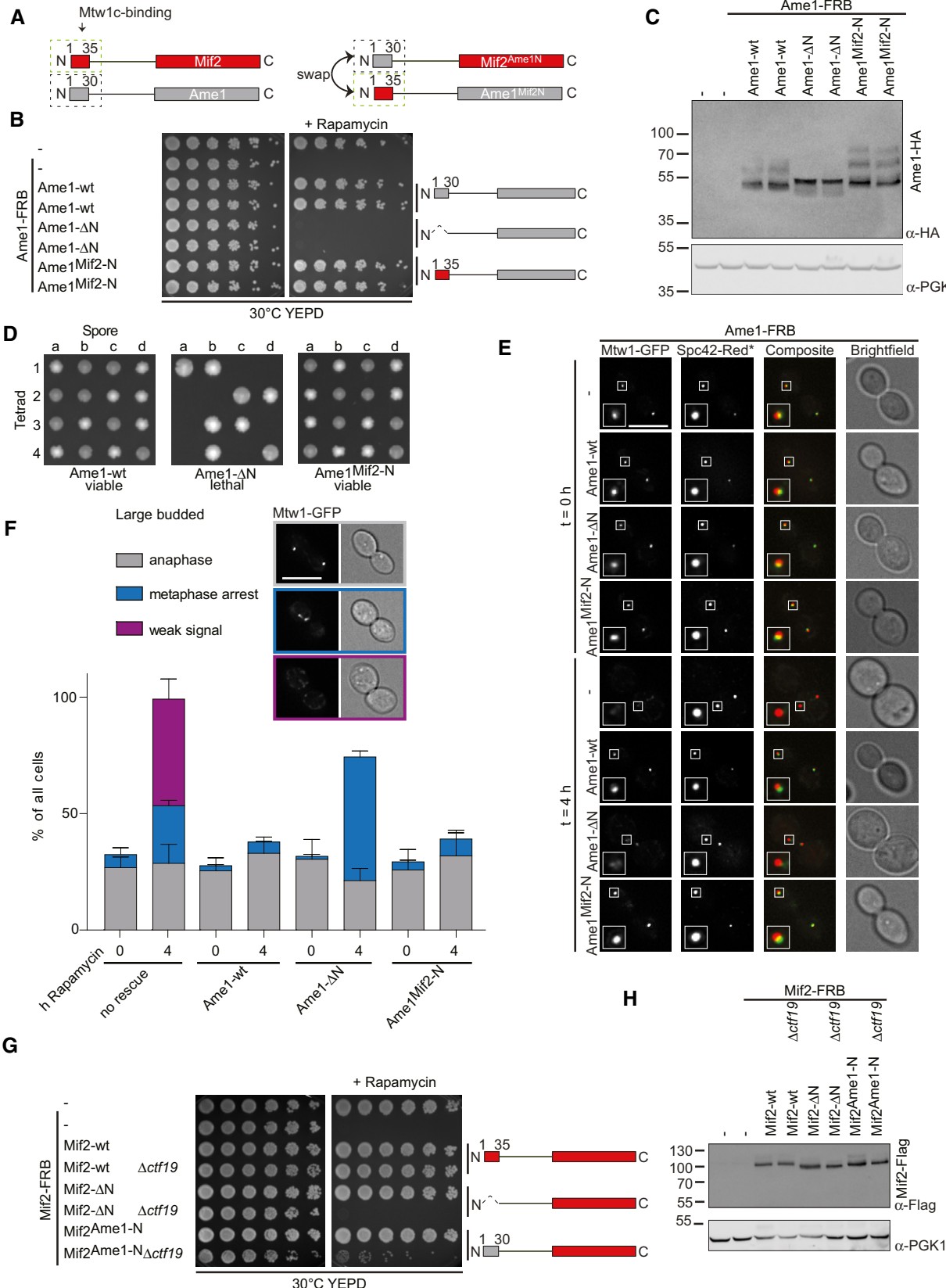

Figure 4.

**Figure 4.  Swap experiments reveal differential functionalities of Ame1 and Mif2N-termini *in vivo*.**

A    Schematic introduction of Ame1 and Mif2 swap mutants in which the N-terminus of one protein is exchanged with the N-terminus of the other.
B    Analysis of Ame1 N-terminal deletion and swap mutant using the anchor-away technique. Ame1-FRB strains containing RPL13-FKBP12 for ribosome anchoring and additionally harboring the indicated rescue alleles were plated in serial dilution on YEPD or YEPD + rapamycin plates incubated at 30°C.
C    Expression levels of 3xHA-tagged Ame1 rescue constructs from (B) analyzed by Western blotting.
D    Tetrad dissection of strains where one endogenous copy of Ame1 was replaced with the indicated construct.
E    Phenotypic analysis of different Ame1 rescue constructs using fluorescence microscopy. Images of representative large-budded cells immediately and 4 h after rapamycin addition are shown, indicated as $t = 0$ h and $t = 4$ h. Bright-field images show the morphology of the corresponding yeast cell. Only large-budded cells were included in the analysis. One kinetochore cluster is enlarged in the white box. Scale bar represents 5 μm.
F    Quantification of (E) in which large-budded cells were classified according to their Mtw1-GFP signal into the three indicated groups. Scale bar represents 5 μm. A total of 100 cells were quantified for every strain. Mean values of the three indicated groups were calculated and plotted in a bar chart. Error bars indicate the SEM of 3 independent experiments.
G    Analysis of Mif2 N-terminal and swap mutant using the anchor-away technique as for (B). Phenotypes of the indicated Mif2 constructs were observed in an otherwise wild-type or *ctf19Δ* background.
H    Western blotting showing 6xFlag-tagged Mif2 expression levels of strains from (G).

This further corroborates the notion that Ame1-N and Mif2-N peptides are functionally equivalent with respect to Mtw1c binding.

## Grafting the Ame1N-terminus onto Mif2 does not provide proper Mif2 function

We next sought to perform the corresponding experiment for Mif2, exchanging its N-terminus with that of Ame1. To evaluate the functionality of the resulting chimeric protein *in vivo*, we first needed to establish conditions in which the N-terminus of Mif2 becomes relevant for cell growth. Using the anchor-away system, we found that in a *ctf19* deletion background, Mif2 wild-type, but not Mif2-ΔN, was able to rescue the lethality of Mif2-FRB in the presence of rapamycin. This effect was not restricted to a *ctf19* deletion, as we also observed lethality in *ctf3* or *chl4* deletion backgrounds, suggesting that the endogenous Mif2 N-terminus becomes important when non-essential budding yeast CCAN subunits are deleted. The *ctf19* deletion background allowed us to assess whether the Ame1N-terminus could function in place of Mif2-N. Interestingly, Mif2$^{Ame1-N}$ chimeras, either containing 15 or 30 amino acids of the N-terminus of Ame1, were unable to efficiently rescue the viability of Mif2-FRB *ctf19Δ* in the presence of rapamycin (Fig 4G), despite being expressed at similar levels as the wild-type protein (Fig 4H). Even in the presence of wild-type Mif2, Mif2$^{Ame1-N}$ displayed compromised growth, suggesting it may have a dominant-negative effect. We confirmed the lethality of Mif2$^{Ame1-N}$ swap in a *ctf19Δ* background by tetrad dissection (Appendix Fig S1). Thus, despite the pronounced similarities between the N-termini, Mif2$^{Ame1-N}$ is functionally not equal to Mif2-wt and cannot provide proper Mif2 function in *ctf19Δ* cells.

## Mtw1c binding by wild-type Mif2 is auto-inhibited and released by Cse4n binding

The genetic analysis indicates that there must be a functional difference between a wild-type Mif2 protein and a variant in which the endogenous N-terminus has been replaced by Ame1-N. As Mtw1c binding by Mif2 becomes important in the *ctf19Δ* background, we speculated that the Mif2$^{Ame1-N}$ swap may either provide insufficient Mtw1 binding activity or may lead to increased or uncontrolled binding. To distinguish between these possibilities, we produced recombinant wild-type Mif2 and Mif2$^{Ame1-N}$ swap in baculovirus-infected Sf9 insect cells and purified the proteins side by side using a carboxy-terminal Flag-tag. Co-infection with Strep-tagged Mtw1

complex indicated that wild-type Mif2 co-purified only little Mtw1c, while the Mif2 swap mutant interacted with a significantly increased amount of Mtw1c as judged by Western blot (Fig 5A). The strongly increased affinity of Mif2$^{Ame1-N}$ swap for Mtw1c was even more apparent in SEC experiments: Upon combination with bacterially expressed Mtw1c, wild-type Mif2 only shifted by a single fraction; in contrast, the Mif2 swap mutant eluted much earlier from the column indicating high-affinity binding to Mtw1c (Figs 5B and EV2).

The most straightforward explanation for Mif2$^{Ame1-N}$ swap displaying an improved interaction with Mtw1c would be that the Ame1N-terminus per se possessed a higher affinity for Mtw1c as compared to its functional counterpart in Mif2. Results from the FP experiments, however, suggest that different binding affinities of the isolated peptides cannot account for the effect seen in Mif2$^{Ame1-N}$ swap, as Ame1 displays a slightly lower binding affinity for Mtw1c (Fig 2B). The binding constant of Ame1 could only be indirectly measured in the FP competition assay, as labeling of Ame1 N-terminal peptides rendered them insoluble. We therefore directly determined binding affinities of Mif2 and Ame1 N-terminal peptides to MN using isothermal titration calorimetry (ITC). With a determined dissociation constant of $K_d = 12.48$ μM, the Ame1-N peptide again displayed a lower affinity for Mtw1c compared to Mif2-N ($K_d = 1.17$ μM). We therefore exclude the possibility that Ame1-N has a higher intrinsic affinity for Mtw1c and that this would cause the effect seen for the Mif2$^{Ame1-N}$ swap mutant (Fig 5C).

Given the results described above, we reasoned that the affinity of Mif2-N for its binding partner MN must be attenuated by some mechanism in the context of the full-length Mif2 molecule. This may occur through the Mif2 N-terminus engaging in intramolecular interactions that would limit the ability to bind MN. To test this idea, we performed comparative chemical cross-linking followed by mass spectrometry of Mif2 either alone or in the presence of a Cse4 nucleosome, the key binding partner of Mif2 at the centromere. Both samples were purified by gel filtration, and the specificity of the distance restraints was increased by brief cross-linking and a short linker length (BS$^2$-Glutarate 7.7 Å). Interestingly, unbound Mif2 displayed many more distance restraints compared to Cse4-bound Mif2 (88 non-redundant cross-links versus 35). Notably, the lysine proximity map indicates long-range intramolecular cross-links between the N-terminal Mtw1c binding domain (represented by Lys9 and Lys12) and residues located close to the Cse4-binding "signature" motif and to putative AT-hook DNA-binding domains in the Mif2 wild-type protein. Out of 10 non-redundant cross-links

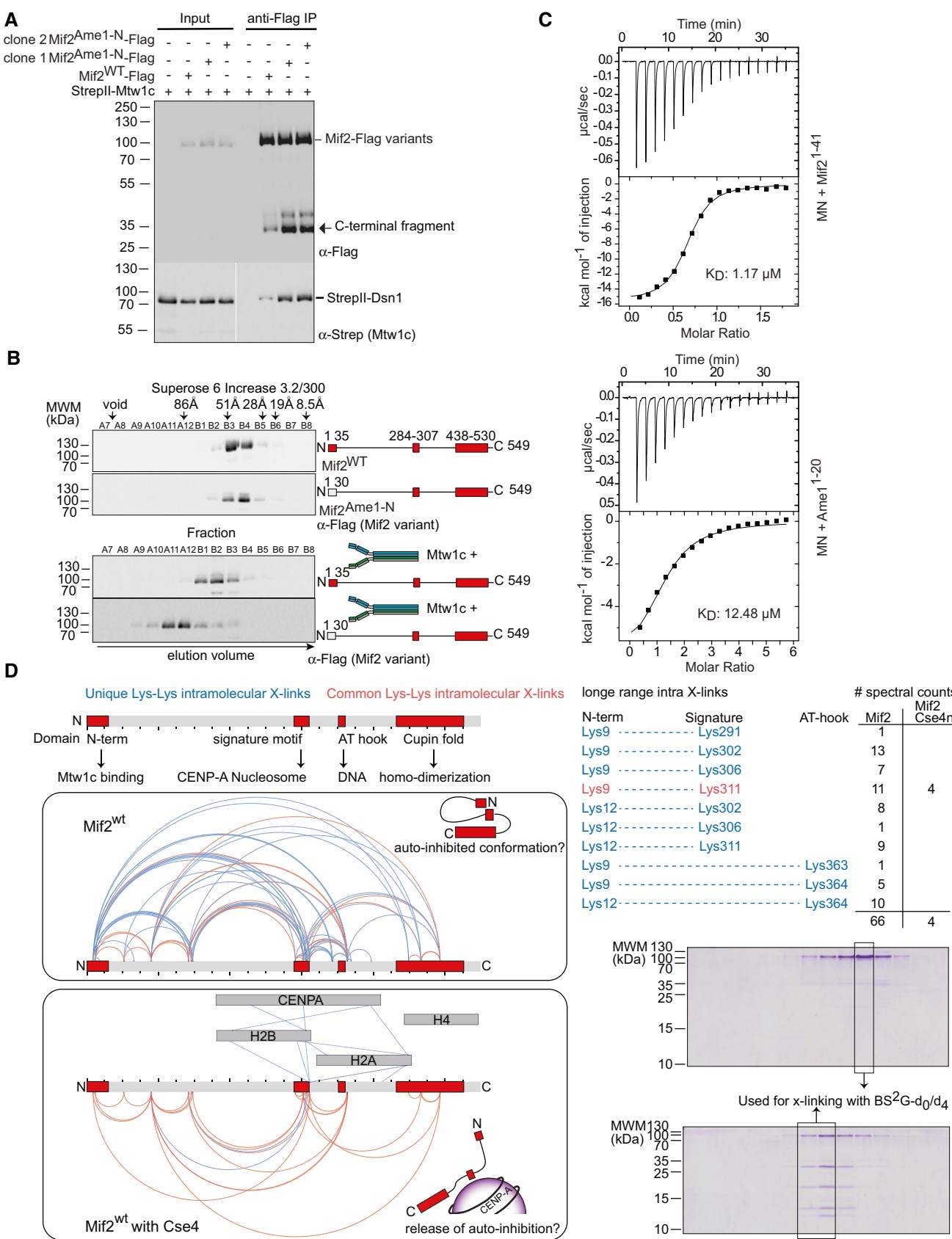

**Figure 5.**

**Figure 5. The Mif2 N-terminus is prevented from high-affinity Mtw1c binding in the full-length molecule.**

A  Co-purification of Strep-Mtw1c through Flag-Mif2 after co-expression in insect cells. Proteins were detected by Western blotting using their respective tags.
B  Western blotting of elution fractions from an SEC experiment of Mif2-wt or Mif2$^{Ame1-N}$ swap in the absence (upper panel) or presence (lower panel) of Mtw1c.
C  ITC analysis of Mif2$^{1-41}$ or Ame1$^{1-20}$ binding to MN. Experiments were performed 3 or 4 times, respectively. One representative binding curve is shown, whereas the $K_D$ value represents the average of all experiments.
D  Cross-link (XL) MS analysis of the Mif2wt protein in isolation or bound to Cse4n using the cross-linker Bis[sulfosuccinimidyl] glutarate. Mif2wt alone or in combination with Cse4n was subjected to analytical size-exclusion chromatography at 10 μM concentration. The corresponding Coomassie-stained SDS–PAGE gels of Mif2-wt alone or in combination with Cse4n are shown. Fractions used for cross-linking are highlighted. Topological map of Mif2 based on the identified intramolecular cross-links. Common cross-links that are found in both samples are shown in red, whereas unique cross-links that can be only found in one of the samples are highlighted in blue. Conserved regions and their described functions are indicated. A list of cross-links that link N-terminal residues to C-terminal regions of Mif2 is shown for Mif2 alone and bound to Cse4n leading to a proposed conformation of Mif2.

Source data are available online for this figure.

between the N-terminus and the signature/AT region, 9 were found exclusively in the Mif2 "alone" sample, while only one (Lys9-Lys311) was found in both samples. Using the number of spectral counts as a semi-quantitative measure for the abundance of cross-links, 66 spectral counts were found for Mif2, but only 4 for Mif2 in complex with Cse4n (Fig 5D). Intermolecular cross-links between the Mif2 signature motif and subunits of the Cse4 nucleosome indicated the validity of the distance restraints detected in this experiment. This raises the possibility that rather than existing as a largely disordered structure, full-length Mif2 may adopt a topology in which the N-terminal Mtw1c binding domain is brought into close proximity to regions involved in nucleosome and DNA binding. This conformation might represent an auto-inhibitory conformation of Mif2, which can be released upon Mif2 binding to Cse4n (Fig 5D).

## Mif2 auto-inhibition involves its Cse4 and DNA-binding motifs

If intramolecular interactions between the Mif2 N-terminus and other parts of the molecule limit the ability to bind MN, then removing this auto-inhibitory domain should increase the affinity for MN. Based on the cross-linking results, we therefore expressed and purified a Mif2 variant that lacks signature motif and AT-hook (Mif2-Δ284–367), as this region displayed the majority of distance restraints with the N-terminus (Fig 6A). Mif2-ΔsigAT was expressed and purified from insect cells in comparable yield and quality to Mif2-wt, indicating stability of this construct (Fig EV3). Consistent with our hypothesis, Mif2-ΔsigAT was able to co-purify a significantly increased amount of MN in solid-phase binding assays as judged by Coomassie staining or Western blotting against the His-tag on Nnf1 (Fig 6B). This result was confirmed in an analytical SEC experiment, in which Mif2-ΔsigAT displayed increased binding to MN as compared to Mif2-wt, as seen in a shift in the chromatogram as well as the shift of the MN in the Coomassie-stained SDS gel (Fig 6C). We conclude that apart from residues in its N-terminus, also residues in the signature motif and AT-hook of Mif2 are responsible for the ability to adopt an auto-inhibited conformation.

## Binding to Cse4 nucleosomes overcomes the inhibited state of wild-type Mif2

To understand how auto-inhibition of the wild-type Mif2 protein may be overcome, we biochemically reconstituted inner kinetochore assembly on centromeric nucleosomes *in vitro*. To this end, we prepared budding yeast mono-nucleosomes containing Cse4, H2A, H2B, and H4 bound to 167 bp of 601 DNA (Cse4n). As expected, wild-

type Mif2 engaged with Cse4n into a stoichiometric complex during SEC (Fig 7A). By contrast, we could not detect any interaction between the Mtw1c and Cse4n under the same conditions (Fig EV4). As shown above, wild-type Mif2 engaged only incompletely with Mtw1c, leading to a shift in elution position by only one fraction (Fig 7B). In the presence of Cse4n, however, Mif2 and Mtw1c readily assembled into a high molecular weight complex with all three components co-eluting in a stoichiometric manner (Fig 7C). Thus, the presence of the Cse4 nucleosome promotes a high-affinity interaction between wild-type Mif2 and Mtw1c. In comparison, binding of Mif2$^{Ame1-N}$ swap to Cse4n was unaltered (Fig 7D), whereas the presence of Cse4n was not required for Mif2$^{Ame1-N}$ swap to engage with Mtw1c (Fig 7E). Mif2$^{Ame1-N}$ swap did not change its elution position in the presence of Mtw1c and Cse4n as compared to in the presence of Mtw1c alone (Fig 7F). Taken together, these biochemical experiments indicate that rather than lacking any key protein–protein interactions, a critical difference between Mif2-wt and Mif2$^{Ame1-N}$ swap mutant is that the latter lacks an auto-inhibitory mechanism that would prevent it from high-affinity binding to Mtw1c in the absence of the Cse4n.

## Expression of Mif2$^{Ame1-N}$ swap leads to chromosome mis-segregation in cells

What is the consequence of expressing Mif2$^{Ame1-N}$ swap in cells? Using the anchor-away system, we noticed that the Mif2$^{Ame1-N}$ swap mutant not only failed to provide viability in the ctf19Δ background upon removing Mif2-FRB from the nucleus, but it also caused reduced growth in the absence of rapamycin, suggesting a dominant effect of this allele (see Fig 4G). To assess this effect in a more systematic manner, we expressed Mif2-wt, Mif2-ΔN, or Mif2$^{Ame1-N}$ swap under the control of a galactose-inducible promoter in an otherwise wild-type background. While overexpression of Mif2-wt and Mif2-ΔN was tolerated well and did not greatly impact yeast viability, expression of Mif2$^{Ame1-N}$ swap from a two-micron plasmid caused lethality on plates (Fig 8A). The toxic effect of this construct was not due to the increased level of the Ame1-N fragment itself, as overexpression of full-length Ame1 alone or together with Okp1 did not cause obvious growth defects (Appendix Fig S2). To further analyze the toxic effect of Mif2$^{Ame1-N}$ swap in cells, we stably integrated the different Mif2 versions under the control of the *GAL10* promoter at the *URA3* locus and again analyzed growth on plates. We included an additional strain in which Mif2 swap was mutated so that it cannot bind Mtw1c (R10E mutation in the RxxG motif). If the growth defect seen in Mif2 swap is a consequence of improper binding to Mtw1c, this effect should be reverted in the R10E mutant,

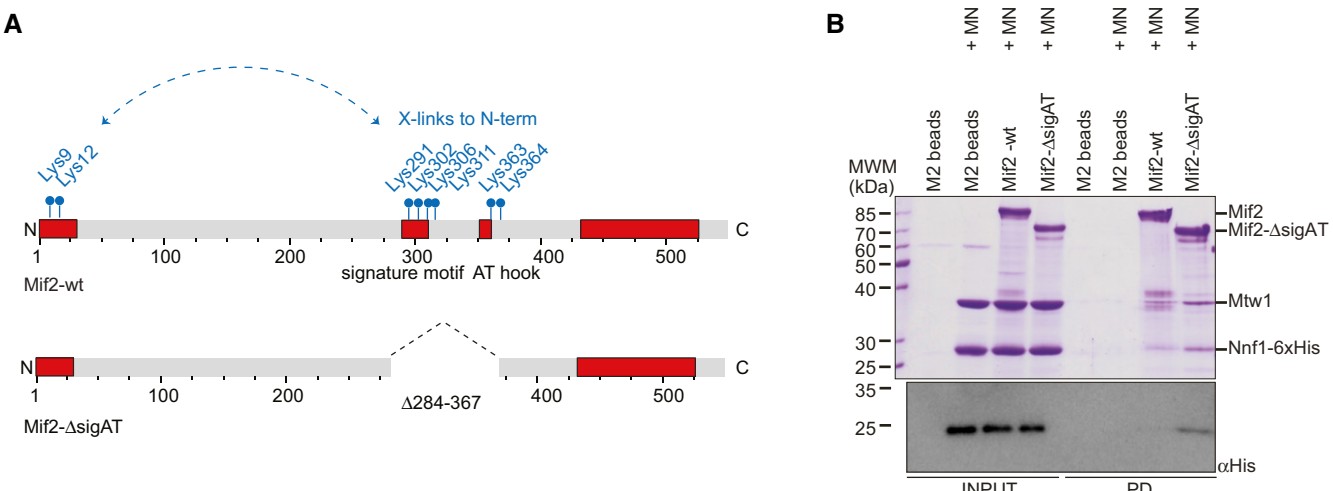

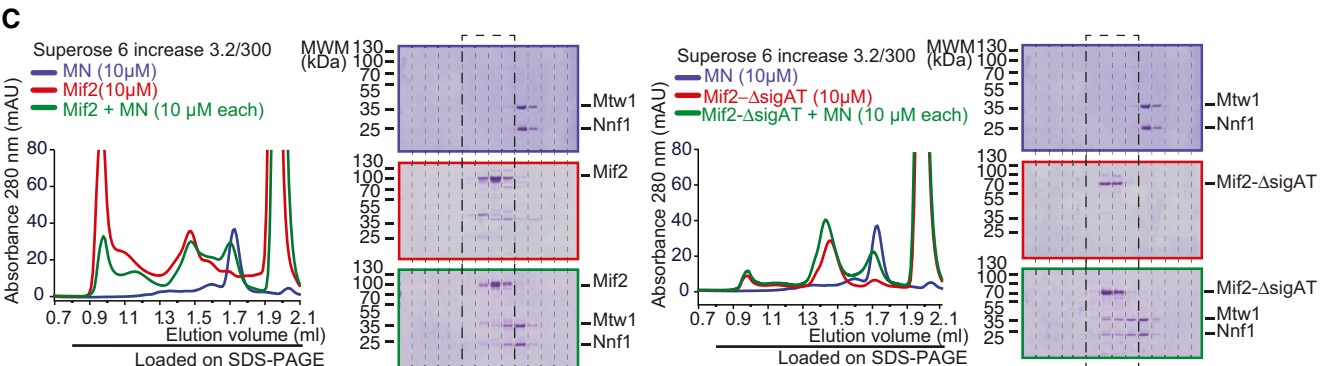

**Figure 6. Mif2 auto-inhibition depends on its Cse4 and DNA-binding motifs.**

A Schematic introduction of Mif2wt and Mif2-ΔsigAT in which its Cse4 and DNA-binding motifs have been removed. Cross-linked lysin residues that connect the N-terminus with residues within the signature motif or AT-hook of Mif2 in isolation are depicted in blue and were the basis to create the Mif2-ΔsigAT mutant.

B Mif2-Flag-wt or Mif2-Flag-ΔsigAT directly from insect cell lysate was immobilized on M2 Flag Agarose, washed, and incubated with Mtw1-Nnf1 complex (MN). After washing, bound complexes were eluted by adding 3xFlag peptide. Western blotting against the His-tag on Nnf1 confirmed increased binding of MN to Mif2-ΔsigAT, already visible in the Coomassie-stained SDS–PAGE gel.

C Analytical SEC chromatograms and corresponding Coomassie-stained SDS–PAGE gels of Mif2-wt or Mif2-ΔsigAT and MN alone and in combination at 10 μM concentration. All combinations were incubated for 1 h at 4°C prior to the run. MN (blue), Mif2 (red), combination (green). Note that the same MN elution profile and SDS–PAGE (upper panel) are displayed in both sets to improve clarity. Dashed boxes highlighting corresponding fractions were included to improve comparability.

which is exactly what we find (Fig 8B). These differential effects on viability were not due to different levels of expression, as all the Mif2 variants were present at similar steady-state levels in yeast extracts, judged by Western blotting (Appendix Fig S3). We labeled chromosome III with GFP and followed chromosome segregation after switching the cells to galactose-containing medium. Fluorescence microscopy revealed that Mif2[Ame1-N] swap cells displayed severe defects in sister chromatid segregation. After overnight growth in galactose-containing medium, 80% of large-budded Mif2[Ame1-N] swap cells displayed either mis-segregation of chromosome III or the signal was entirely absent (Fig 8C). This observation provides an explanation for the pronounced toxicity of the swap construct.

To further define the mechanism by which expression of Mif2[Ame1-N] swap interferes with chromosome segregation, we tagged kinetochores with Mtw1-GFP and spindle pole bodies with Spc42-mCherry in strains that express the different Mif2 variants under a galactose-inducible promoter. Fluorescence microscopy revealed that after switching the cells to galactose, control, Mif2 wild-type, or Mif2-ΔN cells all showed kinetochore signals that were closely associated with spindle pole bodies. By contrast, large-budded Mif2[Ame1-N] swap cells either displayed only scattered Mtw1 fluorescence, or the GFP signal appeared to be stretched between the spindle pole bodies, appearing as multiple dots on short or long spindles, while failing to adopt the typical bi-lobed configuration (Fig 8D and E). These phenotypes were also observed in a mitotic checkpoint-deficient strain background (Fig EV5). The observed chromosome segregation defects might be a direct consequence of increased or inappropriate Mif2[Ame1-N] swap binding to Mtw1c, as observed in vitro (see Fig 5A and B). We thus tested interaction of the different Mif2 constructs with Mtw1c directly in cell extracts in a co-immunoprecipitation (Co-IP) experiment. We could confirm that also in vivo Mif2[Ame1-N] swap more efficiently binds to Mtw1c as

compared to Mif2wt (Fig 8F). This result further predicts that also other Mif2 mutant alleles that are defective in auto-inhibition of Mtw1c binding should have toxic effects. Indeed, we found that expression of the Mif2-ΔsigAT mutant under the control of a galactose-inducible promoter impairs growth, an effect which can be reverted by deleting the Mtw1-binding N-terminus (Fig 8G).

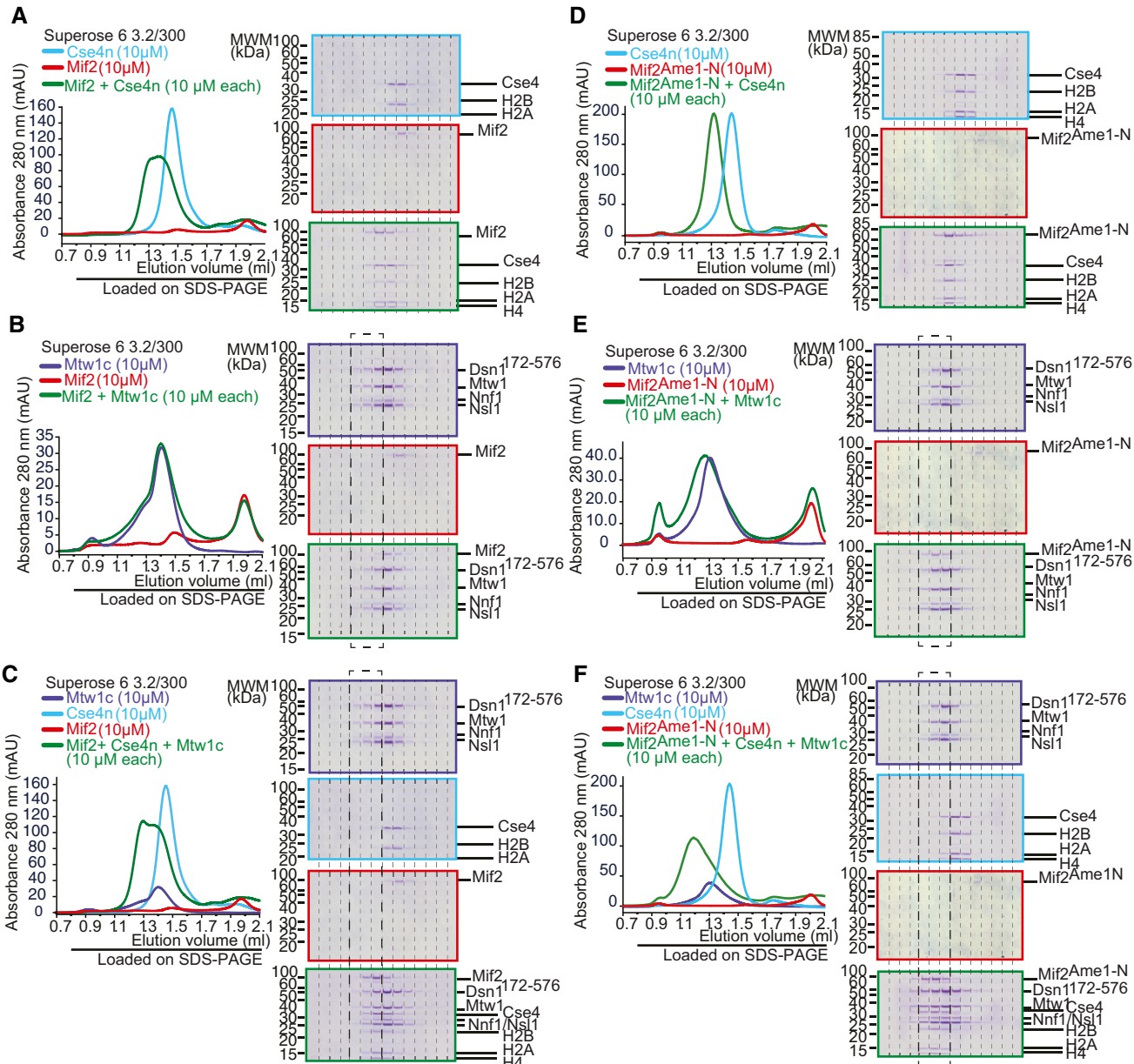

**Figure 7. Mif2 auto-inhibition is released upon binding to a Cse4 nucleosome.**

Analytical SEC chromatograms and corresponding Coomassie-stained SDS–PAGE gels of Mif2-wt or Mif2^Ame-1N swap, Mtw1c, and Cse4n alone and in combination at 10 μM concentration. All combinations were incubated for 1 h at 4°C prior to the run.

A  Cse4n (light blue), Mif2-wt (red), combination (green). Note that the same Cse4n (upper panel) and Mif2 (middle panel) elution profiles and SDS–PAGEs are also displayed as reference in (B, C) (Mif2) or (C) (Cse4n) to improve clarity.

B  Mtw1c (blue), Mif2-wt (red), combination (green). Note that the same Mtw1c (upper panel) elution profile and SDS–PAGE are also displayed as reference in (C) to improve clarity.

C  Mtw1c (blue), Cse4n (light blue), Mif2-wt (red), combination (green).

D  Cse4n (light blue), Mif2^Ame-1N swap (red), combination (green). Note that the same Cse4n (upper panel) and Mif2^Ame-1N swap (middle panel) elution profiles and SDS–PAGEs are also displayed as reference in (E, F) (Mif2^Ame-1N swap) or (F) (Cse4n) to improve clarity.

E  Mtw1c (blue), Mif2^Ame-1N swap (red), combination (green). Note that the same Mtw1c (upper panel) elution profile and SDS–PAGE are also displayed as reference in (F) to improve clarity.

F  Mtw1c (blue), Cse4n (light blue), Mif2^Ame-1N swap (red), combination (green).

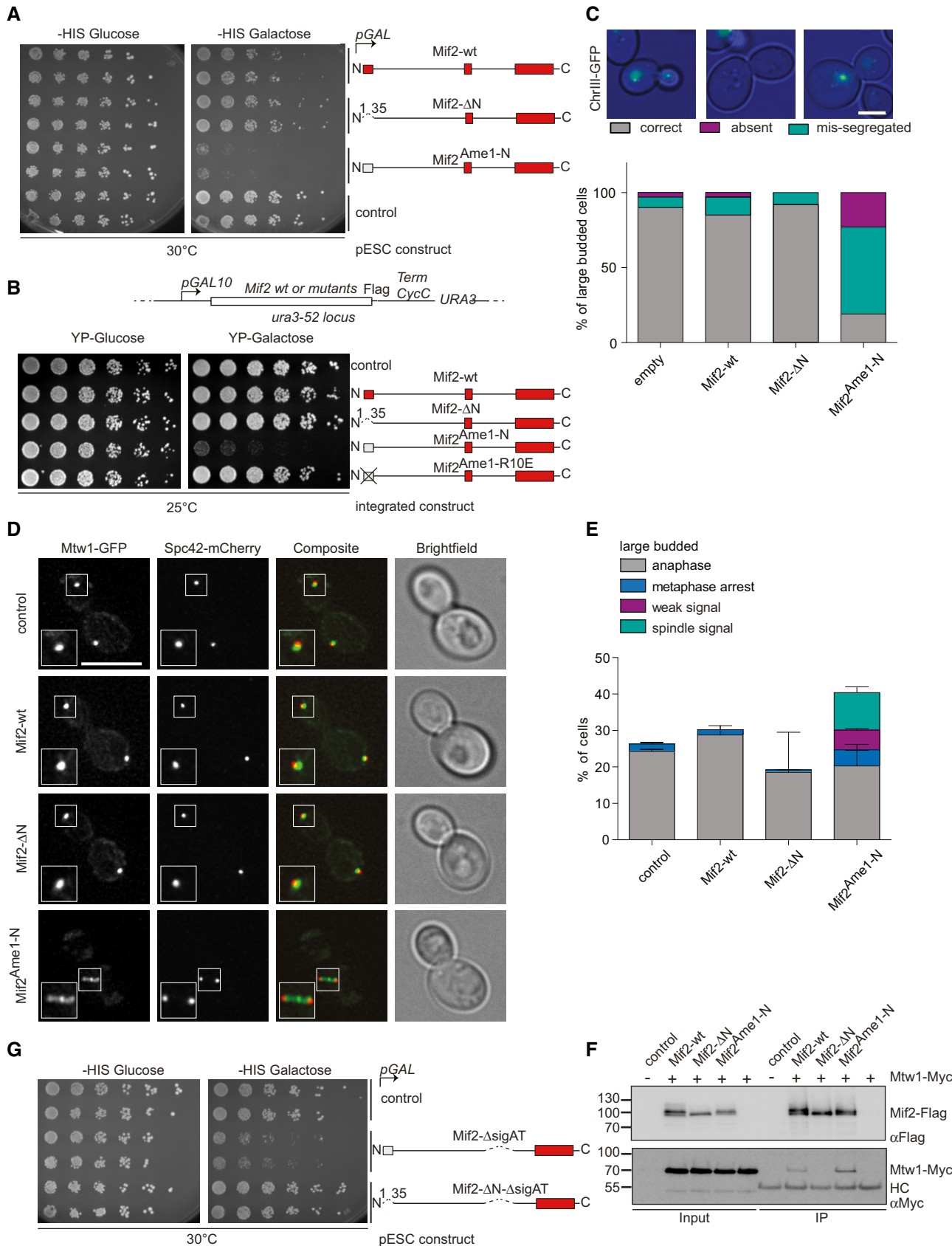

**Figure 8.**

◀

**Figure 8.  Auto-inhibition of Mif2 is crucial for chromosome segregation and cell viability.**

A   Phenotypic analysis of Mif2 mutants using a galactose-inducible overexpression system. Strains harboring the indicated Mif2 expression constructs on pESC plasmids were plated in serial dilution on doHIS-glucose or doHIS-raffinose/galactose plates and incubated at 30°C.

B   Serial dilution assay of chromosomally integrated Mif2 variants under the control of the pGAL promoter. Strains were plated on YEP-glucose and YEP-raffinose/galactose plates and incubated at 25°C.

C   Analysis of chromosome segregation in strains from (B). Distribution of fluorescently labeled chromosome III was analyzed in large-budded cells 16 h after switching the medium to galactose. Scale bar represents 2 μm.

D   Microscopic analysis of kinetochores in Mif2-overexpressing cells visualizing Mtw1-GFP and Spc42-mCherry. Images of representative large-budded cells after 16 h in galactose are shown. One kinetochore cluster is enlarged in the white box for each strain. Scale bar represents 5 μm. Bright-field images show the morphology of the corresponding yeast cell. Only large-budded cells were included in the analysis.

E   Quantification of (D) in which large-budded cells were classified according to their Mtw1-GFP signal into the four indicated groups. A total of 100 cells were quantified for every strain. Mean values of the three indicated groups were calculated and plotted in a bar chart. Error bars indicate the SEM of 2 independent experiments.

F   Co-IP experiment of indicated Mif2-Flag variants and Mtw1-Myc from yeast cell extracts.

G   Phenotypic analysis of Mif2-ΔsigAT mutant using the same galactose-inducible overexpression system as in (A). Strains harboring the indicated Mif2 expression constructs on pESC plasmids were plated in serial dilution on doHIS-glucose or doHIS-raffinose/galactose plates and incubated at 30°C.

## Discussion

Budding yeast kinetochores are assembled from multiple subcomplexes, consist of more than 40 different individual subunits, and eventually comprise hundreds of individual polypeptides (Westermann & Schleiffer, 2013; Musacchio & Desai, 2017). *In vitro*, recombinant kinetochore subcomplexes display significant binding affinity for each other allowing, for example, the biochemical reconstitution of a 13-subunit yeast CCAN or CCAN bound to a nucleosome, whose Cryo-EM structures have been determined recently (Hinshaw & Harrison, 2019; Yan *et al*, 2019). This raises the question as to what may prevent the spontaneous assembly of kinetochores away from centromeric chromatin in cells.

Cooperative binding mechanisms in which multiple low-affinity interactions are combined to drive assembly can contribute to centromere-specific assembly (Weir *et al*, 2016). In addition, centromere-localized regulators such as protein kinases can generate localized signals that promote assembly. In budding yeast for example, centromere-localized Ipl1 kinase contributes to KT assembly through phosphorylation of the Dsn1 subunit of the Mtw1 complex (Akiyoshi *et al*, 2013; Dimitrova *et al*, 2016). While being relevant, these mechanisms alone are probably insufficient to explain the exquisite specificity of kinetochore assembly.

Previous work has characterized CENP-C as a "blueprint" for kinetochore assembly. With binding sites for at least four additional kinetochore subcomplexes organized as non-overlapping linear motifs along the length of its polypeptide chain and its homodimeric organization, it has the potential to act as a powerful nucleator of kinetochore assembly (Screpanti *et al*, 2011; Klare *et al*, 2015; Pentakota *et al*, 2017). Moreover, artificial targeting of the CENP-C N-terminus, along with CENP-T, to non-centromeric loci is sufficient to induce the assembly of ectopic kinetochores (Gascoigne *et al*, 2011; Hori *et al*, 2013). These results raise the question as to how the cell controls CENP-C activity to prevent inappropriate kinetochore assembly. Here, our experiments take advantage of a unique situation that the budding yeast kinetochore architecture offers, in that it incorporates two different "Mtw1 receptors", Mif2 and Ame1. We show that the N-terminal Mtw1-binding peptides of these molecules are, when viewed in isolation, functionally equivalent, demonstrated most clearly by the ability of Ame1[Mif2-N] to rescue the lethality of Ame1-ΔN. This situation is reminiscent of functionally equivalent Ndc80 receptor motifs present in the N-terminus of

Cnn1/CENP-T and in the C-terminus of Dsn1, of which both bind the same binding site on the Spc24-25 heterodimer. Also in this case, the sequence homology between these linear binding motifs is limited, but they do contain a few conserved residues at key positions that are critical to establish similar binding modes (Malvezzi *et al*, 2013; Dimitrova *et al*, 2016).

The analysis of the Mif2[Ame1-N] swap mutant illustrates that despite being similar, the Mtw1c binding motifs can be regulated differently in the context of the respective full-length Ame1 or Mif2 proteins. In a wild-type strain background, Ame1 seems to be the more important binding partner for Mtw1c, as Mif2-ΔN is tolerated, while Ame1-ΔN is not. The reason for this may be that normally more Mtw1c molecules in the kinetochore are connected via Ame1 than via Mif2. In the *ctf19Δ* background, Mtw1c binding by Mif2-N clearly becomes important. The subtle exchange of Mif2-N to Ame1-N in the context of full-length Mif2 yields a Mif2 molecule that now displays a much higher affinity for Mtw1c, bypassing the requirement for the Cse4 nucleosome in the binding of this downstream component in the kinetochore assembly pathway. The severe defects that expression of this mutant causes in yeast cells highlight the importance of Mif2/CENP-C and its regulation in the process of kinetochore assembly. Compared to *S. cerevisiae*, proper regulation of CENP-C will be even more important in organisms that exclusively rely on CENP-C to connect centromeric chromatin to the KMN network of the kinetochore, such as *C. elegans* or *D. melanogaster*. We currently do not know whether auto-inhibition plays any role in regulating the Ame1-Mtw1c interaction, as well. At least in a wild-type strain background, the Ame1[Mif2-N] swap mutant did not create any phenotype. The mode of Cse4 binding differs between Ame1-Okp1 and Mif2, suggesting that the described mechanism of inhibition release may be specific to Mif2 (Anedchenko *et al*, 2019; Fischböck-Halwachs *et al*, 2019). In the context of kinetochore assembly, auto-regulation of Mif2 may be more critical, because this subunit has the higher potential to bind additional kinetochore components and titrate them away from the kinetochore. In addition, post-translational modifications such as phosphorylation of Dsn1 by Ipl1 allow additional levels of regulation (Dimitrova *et al*, 2016).

According to our findings, we propose a model in which Mif2 serves as regulatory platform for kinetochore assembly (Fig 9). Its auto-inhibited conformation prevents its interaction with the Mtw1 complex in the nucleoplasm. Only when bound to the centromere

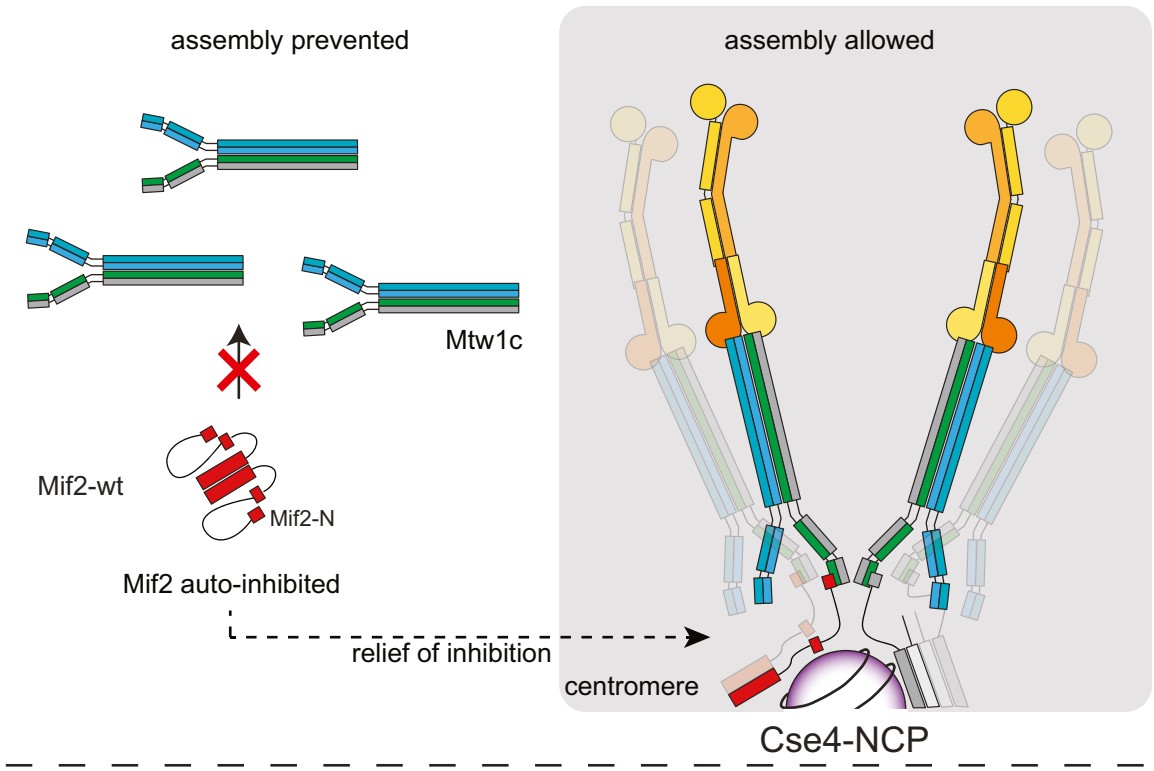

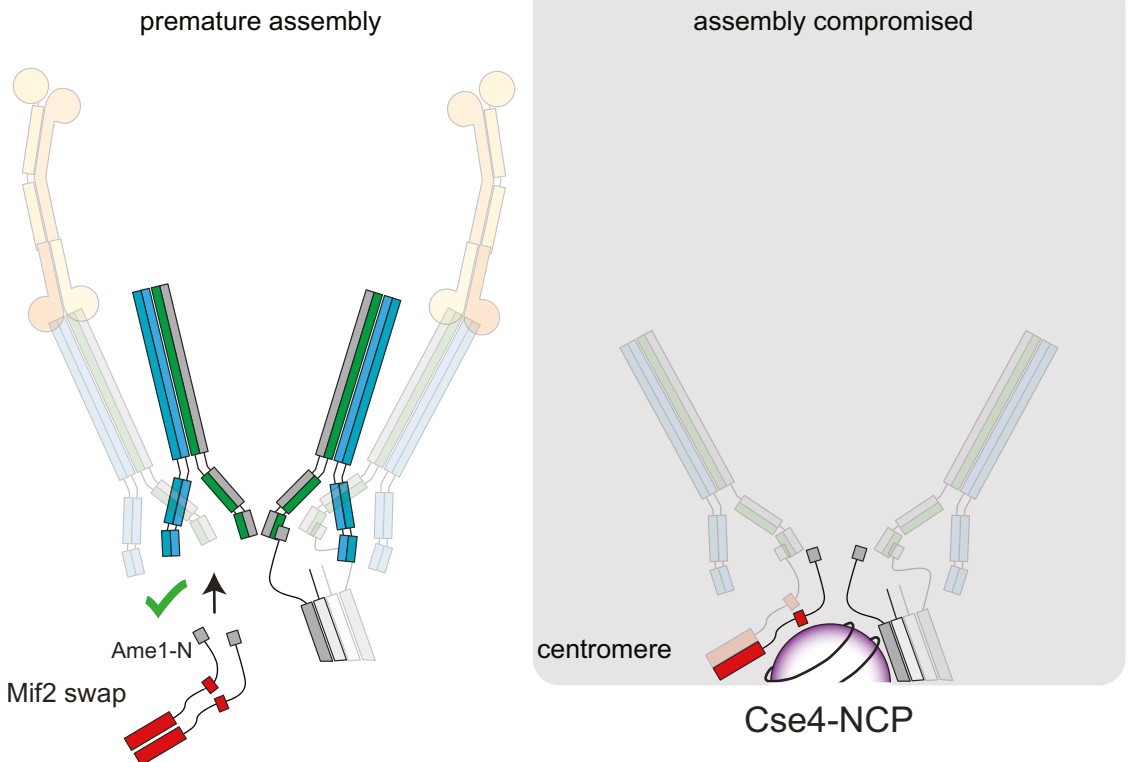

**Figure 9. Model for intrinsic regulation of Mif2 during kinetochore assembly.**

Upper panel: In a wild-type situation, Mif2 does not bind to the Mtw1 complex in the nucleoplasm as a consequence of adopting an auto-inhibitory conformation. Binding of Mif2 to the Cse4 nucleosome allows a conformational change that promotes its interaction with the Mtw1c and thus triggers full kinetochore assembly. Lower panel: Exchanging the N-terminus of Mif2 with the N-terminus of Ame1 prevents the adoption of an auto-inhibitory conformation. As a consequence, Mif2 forms interactions with the Mtw1c already in the nucleoplasm. These interactions might promote the assembly of soluble kinetochores that are able to bind to microtubules or chromatin away from centromeres and are therefore not available for proper kinetochore assembly on Cse4 nucleosomes.

through a direct interaction with Cse4 nucleosomes, Mif2 releases its auto-inhibitory conformation and allows formation of a functional kinetochore (Fig 9 upper panel). By exchanging the N-terminus of Mif2 with the N-terminus of Ame1 or by deletion of Mif2 signature motif and AT-hook, the inhibitory conformation of Mif2 is prematurely released, allowing formation of soluble kinetochores which interfere with proper chromosome segregation (Fig 9 lower panel).

We propose that Mif2[Ame1-N] swap exerts its toxic effect by preventing the Mif2 molecule from adopting an auto-inhibited conformation, rather than simply from an enhanced affinity of the Ame1-N versus the Mif2-N fragment for Mtw1c itself. The following three observations argue in favor of this model: (i) We showed that the dissociation constants of Mif2-N and Ame1-N peptides for Mtw1 constructs are similar and both in the low micromolar range. In these assays, Mif2-N even appeared to have a higher affinity for MN compared to Ame1-N, making it even more surprising that the wild-type full-length Mif2 molecule is such a weak Mtw1c binder. (ii) We show in this study that Mif2-N can replace Ame1-N in the context of the Ame1 molecule; this ability to rescue would be difficult to explain if Mif2-N had a greatly reduced affinity for Mtw1c compared to Ame1-N. (iii) A mutation that does not alter the Mif2 N-terminus itself, but a putative binding region for Mif2-N within the Mif2 protein (Mif2-ΔsigAT), also leads to increased binding of MN and reduced viability in cells. Controlling Mif2 conformation and releasing auto-inhibitory conformations might also be exploited by kinases in the process of kinetochore assembly regulation. We note that Mif2 is an Ipl1 substrate and that non-phosphorylatable Mif2 mutants cause chromosome segregation defects (Westermann *et al*, 2003). Future work will have to define how the Mif2 conformation can be regulated by post-translational modifications.

The existence of auto-inhibited conformations of kinetochore molecules, which can be released during the assembly process, appears to be an emerging molecular topic. In addition to the well-characterized inhibitory effect of the Dsn1N-terminus, which can be released through Ipl1 phosphorylation to promote Mif2 binding (Dimitrova *et al*, 2016) in a mechanism that is conserved in other eukaryotes (Bonner *et al*, 2019), a recent study has provided evidence for an intrinsic regulation of Ndc80's microtubule binding activity through the adoption of a tightly folded conformation (Scarborough *et al*, 2019). It will be interesting to explore whether similar strategies are employed by even more kinetochore subunits. The architecture of kinetochore subcomplexes, which often combine relatively short structured domains with long, intrinsically unstructured segments, makes this strategy particularly feasible to control kinetochore assembly.

While shortcutting the regular kinetochore assembly pathway and being detrimental for the cell, the Mif2[Ame1-N] allele we introduce in this study might be useful for the future biochemical and structural analysis of kinetochores. Pioneering studies have affinity-purified yeast kinetochores for initial structural and functional experiments (Akiyoshi *et al*, 2010; Gonen *et al*, 2012; Gupta *et al*, 2018). These approaches, however, suffer from relatively low yield and homogeneity of the isolated kinetochores, limiting their usage. Genetic tools such as the Mif2[Ame1-N] swap mutant, potentially in combination with other alleles, might be valuable to improve the solubility and accessibility of kinetochores from yeast extracts in order to facilitate biochemical and structural analysis of native kinetochores.

# Materials and Methods

**Production of recombinant proteins**

Expression constructs for kinetochore proteins used in this study were created by amplification of the DNA for the respective genes from yeast genomic DNA and cloning into pETDuet-1, pET3aTr/pST39, pST44, or pGEX6P plasmids (bacterial expression) or pFL plasmids (insect cell expression) following the protocol for restriction-free cloning. Restriction-free cloning was also used to produce vectors encoding truncated and fusion proteins. Site-directed mutagenesis (Agilent Technologies) was applied for introduction of amino acid substitutions. A list of all vectors used for protein production and purification in bacteria or insect cells can be found in Appendix Table S2.

Following conditions were used for protein production and purification unless indicated otherwise: Competent bacterial cells were grown at 37°C until OD600 of 0.6 and subsequently induced with 0.5 mM IPTG. Expressions were conducted overnight at 18–20°C with the exception of the MN and Cse4 histone octamer (0.5 mM or 1M IPTG, respectively; 4 h at 37°C). Expression of Mtw1c, MN, and GST-Mif2 or GST-Ame1 constructs was performed in BL21 (DE3; Novagen), AOc was expressed in Rosetta 2(pLys) (DE3) cells (EMD Millipore) and Cse4 histone octamer in BL-21-CodonPlus(DE3)-RIL, whereas Mif2 was expressed and purified from Sf9 insect cells. Lysis, wash, and elution buffers as well as chromatography steps varied for the different protein complexes and are described in the individual sections. The poly-histidine fusion proteins were isolated with HisTrap HP 5-ml columns, and GST-tagged proteins were purified with use of glutathione Sepharose 4B slurry (GE Healthcare). In the last step of all bacterially expressed proteins, the protein was passed over a size-exclusion chromatography (SEC) column that was appropriate for the protein size while elution of proteins was measured by absorbance at 280 nm using an Akta FPLC System connected to a Windows-based laptop running the UNICORN control software. Mif2 was expressed in insect cells and purified in batch using M2 affinity agarose (Sigma-Aldrich). All proteins were concentrated to the desired concentration and flash-frozen in liquid nitrogen before stored at −80°C until usage. None of the proteins was reused after thawing. For pulldown experiments on Mif2-Flag, proteins were not eluted from M2 agarose beads after washing, but directly used for incubation with MN.

### Mtw1c

The Mtw1 complex (Mtw1c) used in this study includes an N-terminal 171 amino acid truncation of the Dsn1 subunit and an N-terminal 6xHis-tag at the same protein. Lysis buffer for Mtw1c was 50 mM Tris–HCl, pH 7.5, 500 mM NaCl, 30 mM imidazole, 10% glycerol, and 5 mM β-mercaptoethanol as described previously (Maskell *et al*, 2010). Lysates were loaded onto a HisTrap HP 5-ml column pre-equilibrated in lysis buffer, washed with 30 column volumes (CV) lysis buffer, and eluted with 30 mM Tris, pH 8.5, 80 mM NaCl, 10% glycerol, 5 mM β-mercaptoethanol, and 250 mM imidazole. Subsequently, proteins were directly loaded onto anion exchange chromatography (HiTrap Q HP 5-ml column; GE Healthcare). The column was developed with 4 CV gradient wash from 0 to 20% buffer B, followed by 15 CV wash of 20% buffer B. The chromatography was performed with a gradient consisting of buffer (30 mM Tris–HCl, pH 8.5, and 5% glycerol from 80 mM [A] to 1 M

NaCl [B]) using a flow rate of 1 ml/min and an elution volume of 40 CV. The protein is eluted at a buffer B ratio of about 40%. Fractions containing the Mtw1c were concentrated using a Vivaspin 20 ultracentrifugal unit MWCO 10000 Dalton (Sartorius) and loaded to a HiLoad Superdex 200 16/600 pg (GE Healthcare) equilibrated in 30 mM HEPES, pH 7.5, 250 mM NaCl, 5% glycerol, and 2 mM TCEP. All Mtw1c amino acid substitutions used in this study were expressed and purified according to the same protocol.

### Mtw1/Nnf1

In the Mtw1/Nnf1 dimer used in this study, a 6xHistidine-tag was introduced N-terminal of Nnf1. Purification was performed according to the protocol for Mtw1c described here, except that the dimer did not bind to the HiTrap Q HP 5-ml column. Therefore, the flow-through of the column was used for concentration and subsequent loading to the Superdex 200 10/300 GL column.

### AOc

Bacterial pellets expressing the AO complex (AO) were resuspended in lysis buffer (30 mM HEPES, pH 7.5, 30 mM imidazole, 600 mM NaCl, and 5 mM β-mercaptoethanol). AO loaded on the HisTrap HP 5-ml column was washed with 30 CV lysis buffer before eluted with 300 mM imidazole in the otherwise same buffer. Subsequently, the complex was purified over gel filtration Superdex 200 10/300 GL or HiLoad Superdex 200 16/600 pg columns (GE Healthcare) in 30 mM HEPES, pH 7.5, 250 mM NaCl, 5% glycerol, and 2 mM TCEP. The complex eluted in two peaks from the column, of which the later eluting peak was used for subsequent experiments.

### Mif2

The open reading frame encoding Mif2 was amplified from yeast genomic DNA and cloned into the pFL vector for insect cell expression. A 1xFlag-tag was added C-terminal to the protein. Viruses expressing the protein were produced according to the multi-Bac system (Trowitzsch *et al*, 2010). For purification, insect cells were opened with a Dounce homogenizer in lysis buffer (50 mM HEPES, pH 7.5, 500 mM NaCl, and 5% glycerol). Cleared extracts were incubated with M2 anti-Flag affinity agarose (Sigma-Aldrich) for 2 h and washed with 20 CV lysis buffer and 30 CV wash buffer (20 mM HEPES, pH 7.5, 150 mM NaCl, and 2.5% glycerol) before eluted in the same buffer containing 200 μg/ml 3xFlag peptide for 1 h rotating at 4°C. The protein was subjected to a Vivaspin 500 ultracentrifugal unit MWCO 5000 Dalton (Sartorius) and concentrated to the desired concentration. Mif2 variants were produced according to the same protocol.

### GST constructs

GST-Mif2_1-41 and GST-Ame1_1-30 bacterial expression cultures were lysed in 50 mM HEPES, pH 7.5, 300 mM NaCl, 5% glycerol, and 5 mM β-mercaptoethanol before incubated with glutathione Sepharose 4B slurry at 4°C while rotating. Protein-loaded beads were subjected to a disposable gravity flow column (Bio-Rad), washed with 40 CV of lysis buffer, and eluted in 3-ml steps with lysis buffer supplemented with 20 mM reduced glutathione. The first three fractions that included the protein were concentrated in a Vivaspin 20 ultracentrifugal unit MWCO 10,000 Dalton and subjected to a Superdex 200 10/300 GL or HiLoad Superdex 200 16/600 pg column.

### Cse4 nucleosomes

A cDNA encoding for all histones of the Cse4 nucleosome (Cse4, H4, H2A, and H2B; including a C-terminal Flag and His-tag on the H2B subunit) was cloned into a polycistronic expression vector, co-expressed in *E. coli,* and purified. Purification included lysis of bacterial pellets in lysis buffer (50 mM HEPES 7.5, 1 M NaCl, and 1.5 mM $MgCl_2$, 50 mM imidazole) and loading of cleared lysate to a HisTrap HP 5-ml column before washing with 30 CV of lysis buffer. Histone proteins were eluted in lysis buffer supplemented with 500 mM imidazole in a one-step gradient. As peak fractions contain an excess of H2A/H2B dimer and a suboptimal buffer condition for the following nucleosome reconstitution, the sample was run over a HiLoad Superdex 200 16/600 pg SEC column (2 M NaCl, 10 mM Tris–HCl, pH 7.4, 1 mM EDTA). The early eluting peak included all histone proteins, whereas the later eluting peak represented the excess of H2A/H2B dimer. Histone proteins can be flash-frozen and stored at −80°C.

A pUC19 plasmid including 16 repeats of the 601-Widom DNA-167 bp sequence (Lowary & Widom, 1998) has been amplified in XL10-Gold Ultracompetent Cells and extracted with a NucleoBond PC 10000 Kit (Macherey-Nagel) according to the manufacturer's protocol. 4 mg of the isolated plasmid was subjected to cleavage with ScaI-HF (NEB), which releases all 16 inserts from the vector backbone. To separate the insert from the vector, the vector was precipitated by adding 770 mM NaCl and 14% PEG 6000 final concentrations and centrifugation at 27,000 *g* for 20 min at 4°C. The supernatant including the 601-Widom DNA was precipitated by resuspending the pellet in TE10/0.1 (10 mM Tris–HCl 8.0, 0.1 mM EDTA), adding 30% EtOH final concentration, and incubating at −20°C overnight before spinning at 3,200 *g* at 4°C for 1 h. DNA pellets were dried completely before resuspended in TE10/0.1 and adjusted to a final concentration of 0.3 mg/ml.

The reconstitution of Cse4 nucleosomes, consisting of the purified histones and 601-Widom DNA, was accomplished with slight modifications in accordance with a protocol from the prior art (Guse *et al*, 2012). Very briefly, histones and DNA were mixed and dialyzed in a two-step dialysis to a final low salt buffer (50 mM NaCl, 10 mM TEA, pH 7.4, 1 mM EDTA). Reconstituted nucleosomes were stored at −80°C.

## Yeast genetics

Yeast strains were constructed in the S288C background. A list of all yeast strains used in this study can be found in Appendix Table S3, and a list of all vectors used for generation of novel yeast strains can be found in Appendix Table S2. Yeast strain generation and methods were performed by standard procedures (Daniel *et al*, 2006).

The anchor-away approach for characterization of Ame1, Mif2, and Mtw1 mutants was performed as described (Haruki *et al*, 2008), using the ribosomal RPL13-FKBP12 anchor. Final rapamycin concentration in plates or liquid media was 1 μg/ml.

For live-cell microscopy of strains expressing different variants of Ame1 in an Ame1-FRB background, cells were grown in YEPD medium supplemented with 1 μg/ml rapamycin for 4 h and images were taken before and after addition of rapamycin. Microscopy of strains overexpressing different variants of Mif2 upon induction with galactose was grown overnight in the presence of 2% raffinose

and galactose, diluted to an OD600 of 0.3, and grown for 4 h before images were taken in the presence of galactose. Images were taken in synthetic medium without tryptophan with either glucose (Ame1 strains) or raffinose/galactose (Mif2 strains) as sugar source. Strains were imaged on concanavalin A-coated glass-bottom culture dishes (No. 1.5 coverglass, MatTek Corporation) by live-cell DeltaVision Deconvolution Microscopy on a DeltaVision Elite System (GE Healthcare) controlled by Softworx and equipped with a Plan Apochromat 100× or 60× 1.4 NA objective (Olympus) and an EDGE sCMOS camera at 25°C. Z stacks (13 × 0.4 µm apart) were acquired and projected into 2D images. Images were processed and analyzed using ImageJ and Photoshop (Adobe). Large-budded cells were subclassified into different categories (metaphase, anaphase arrest, Mtw1-GFP signal loss, and Mtw1-GFP signal on the spindle) visually by considering bud size, Spc42-mCherry signal distribution, Mtw1-GFP signal intensity, and distribution. For each strain of each experiment, at least 100 cells were classified into two to three independent biological experiments.

To visualize yeast chromosome III, expression of pCu-LacI-GFP in strains carrying a lacO array inserted 23 kb away from CENIII was induced by adding $CuSO_4$ to a final concentration of 250 µM to the medium prior to imaging. 20 z-sections with 0.4-µm spacing were acquired and projected into 2D images.

Serial fourfold dilutions of overnight cultures were prepared on 96-well plates in minimal medium starting from OD600 of 0.6. The dilutions were spotted on YPD medium with and without rapamycin or on YP-glucose and YP-galactose and grown at 30°C for 3 days. To confirm phenotypes observed in the serial dilution assays for Mtw1 mutants, Mtw1 hemizygous deletion strains were created, and Mtw1-wt or mutants were integrated at an exogenous locus before haploid spores were produced. To confirm phenotypes observed for Ame1, one copy of Ame1 in a diploid strain was replaced by wild-type or mutant Ame1 and the recovery of haploid spores was investigated. Expression of integrated proteins was checked for all created yeast strains by protein extraction from yeast (Kushnirov, 2000) and Western blotting against the respective tags of individual proteins (see Appendix Table S1).

## Multiple sequence alignments

Sequence homologs were collected within the NCBI non-redundant protein database, using NCBI-BLAST (version 2.2.24, E-values 0.01) (Altschul *et al*, 1997). Sequence accession numbers are listed in Appendix Table S4. The alignments were visualized and processed with Jalview (Tcoffee alignment, Clustalx coloring scheme with 30% conservation threshold) (Waterhouse *et al*, 2009).

## Co-Immunoprecipitation

80 ml of yeast culture at $OD_{600}$ of 0.6 was harvested and resuspended in lysis buffer (20 mM Hepes pH 7.5, 300 mM NaCl, 2.5% glycerol, 2 mM TCEP, 0.05% Triton X-100, 10× protease inhibitors (Thermo Fisher)) and disrupted in a Mini-Beadbeater (BioSpec Products) at 4°C. Extracts were cleared at 15K for 20 min at 4°C. 1 mg of lysate was loaded on M2 affinity agarose (Sigma-Aldrich) which was preblocked with 5 mg/ml of BSA. After incubation for 3 h at 4°C while rotating, samples were washed 4 times for 5 min in lysis buffer without protease inhibitor. Proteins were eluted from beads

via the addition of 2 × Laemmli sample buffer, and samples were run on an SDS–PAGE gel and analyzed in Western blotting with α-Flag and α-Myc antibodies.

## Biochemical interaction studies

### GST and Flag pulldown experiment

The GST pulldown experiment was performed using BSA-preblocked GSH Sepharose beads. Flag pulldown experiments used M2 Flag agarose beads. GST, GST-Mif2_1-41 and GST-Ame1_1-30, or empty beads, Mif2wt, or Mif2-ΔsigAT coupled beads served as bait at 1 µM, whereas MN served as prey at 3 µM concentration. For competition experiments, Ame1_1-20 peptide dissolved in PBS served as competitor at the indicated concentrations. Proteins were incubated on the beads in the presence or absence of peptide for 1 h at 4°C in pulldown buffer (10 mM HEPES, pH 7.5, 200 mM NaCl, 0.05% Triton, 2.5% glycerol, 1 mM TCEP). Beads were washed twice in pulldown buffer, before Laemmli sample buffer was added to the beads (GST pulldown) or proteins were eluted in pulldown buffer supplemented with 1 mg/ml Flag peptide before addition of Laemmli sample buffer (Flag pulldown) and samples were analyzed on a Coomassie Brilliant Blue-stained SDS–PAGE gel or Western blotting. Proteins for analytical SEC experiments were mixed in 50 µl final volume in equimolar ratios and loaded onto Superose 6 3.2/30 or Superose 6 Increase 3.2/30 columns. All SEC interaction studies were conducted under isocratic elution conditions at 4°C in SEC buffer (20 mM HEPES, pH 7.5, 150 mM NaCl, 2.5% glycerol, 1 mM TCEP or 10 mM HEPES, pH 7.5, 50 mM NaCl, and 1 mM TCEP for experiments including Cse4 nucleosomes) on an Ettan LC System (GE Healthcare). Elution of proteins was monitored at 280 nm. 100 µl fractions were collected and analyzed by SDS–PAGE and Coomassie Brilliant Blue staining or Western blotting. The curves for 280-nm absorption were overlaid in UNICORN software (GE Healthcare). To test co-purification from baculovirus-infected Sf9 cells (Fig 5), pellets were resuspended in lysis buffer consisting of 20 mM $Na_2HPO_4$,/$NaH_2PO_4$, pH 6.8, 300 mM NaCl, 2.5% glycerol, and 0.05% Tween-20. Cells were lysed by sonication, and cleared extracts were incubated with M2 affinity agarose resin (Sigma-Aldrich). After several washes with lysis buffer, proteins were eluted with 0.2 mg/ml 3xFlag peptide in lysis buffer, and fractions were analyzed by Western blotting. SEC interaction studies shown in Figs 5 and EV2 were performed in 25 mM HEPES-KOH, pH 7.5, 200 mM NaCl, 1 mM $MgCl_2$, 2% glycerol, and 0.5 mM TCEP.

### Fluorescence polarization (FP) assay

FP experiments were performed in 30 mM Hepes pH 7.5, 150 mM NaCl, and 1 mM TCEP at 23°C with a JASCO FP-8300 fluorescence spectrometer and the software Spectra Manager. Data analysis was done with GraphPad Prism 5 assuming one set of binding sites. Each experiment was repeated three times and the mean with standard deviation represented in the graphs. Titration fluorescence polarization measurements were performed with an excitation wavelength of 490 nm and an emission wavelength of 517 nm. To measure FITC-Mif2[1-41] binding to MN, each reaction contained 0.1 µM FITC-Mif2 and MN was added in increasing concentrations, ranging from 0 to 4 µM.

For Ame1[1-20] competition experiment, each reaction contained 0.1 μM FITC-Mif2[1-41], 0.4 μM MN, and increasing concentration of Ame1-N ranging from 0 to 40 μM. The IC50 (which is the half-maximal reduction of "r") in a competition titration is defined as follows: Log IC50 = log/10^logK$_i$(Ame1_1-20)(1 + conc(FITC-Mif2)/K$_D$(FITC-Mif2)) with r = Bottom + (Top-Bottom)/(1 + 10^(conc(Ame1_1-20)-log IC50) where Top and Bottom describe the upper and lower plateau of the binding curve. The IC50 is dependent on the K$_i$ of Ame1 peptide, the concentration of FITC-Mif2, and the K$_D$ of FITC-Mif2_MNc.

## Isothermal titration calorimetry (ITC) experiment

ITC measurements were performed in 30 mM Hepes pH 7.5, 150 mM NaCl, and 1 mM TCEP at 25°C using a MicroCal ITC200 System. Mif2_1-41 (250 μM) and Ame1_1-20 (800 μM) were titrated in 2-μl steps (18 in total) into the device cell containing 25 μM MN. Thermodynamic values and stoichiometries were obtained using Origin 7 (OriginLab Corp.) by fitting the binding isotherms with the non-linear least squares method, assuming one set of binding sites.

## Chemical cross-linking and MS of Mif2wt and Mif2-Cse4n complex

Prior to cross-linking, samples were subjected to SEC Superose 6 3.2/300 column in 20 mM Hepes 7.5, 50 mM NaCl, and 1 mM TCEP. Peak fractions of Mif2 or Mif2-Cse4n complex were cross-linked with 600 μM isotopically labeled (d$_0$/d$_4$) Bis[sulfosuccinimidyl]glutarate (Thermo Scientific) at 35°C for 5 min. The reaction was quenched with 100 mM ammonium bicarbonate.

## Data availability

No datasets were deposited in public repositories.

**Expanded View** for this article is available online.

## Acknowledgements
The authors wish to thank Doro Vogt and Andrea Musacchio (MPI Dortmund) for advice and practical help with nucleosome purification. We thank Christine Beuck and Melisa Merdanovic for advice on biophysical assays. We thank Alwin Köhler (University of Vienna) for expression plasmids, Ivana Primorac for help with PyMOL illustrations, and all members of the Westermann laboratory for discussions. We thank the Imaging Center Campus Essen (ICCE) for support with microscopy. S.W., M.B., and P.B. acknowledge support from the Collaborative Research Center CRC 1093 Supramolecular Chemistry on Proteins. This work received support from the German Research Foundation (DFG), grant no. WE-2886/2.

## Author contributions
KK performed biochemical and genetic experiments, supported by MBö, PS, SH, SW, and KJ. MBl performed and analyzed FP experiments, supervised by PB. GH and FH performed and analyzed XL-MS experiments. KK and SW designed research, analyzed data, and wrote the manuscript.

## Conflict of interest
The authors declare that they have no conflict of interest.

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
