## [Review Process File · The EMBO Journal]

Auto-inhibition of Mif2/CENP-C ensures centromere-dependent kinetochore assembly in budding yeast

Stefan Westermann, Kerstin Killinger, Miriam Böhm, Philine Steinbach, Karolin Jänen, Simone Hohoff, Goetz Hagemann, Franz Herzog, Mike Blüggel, and Peter Bayer

DOI: N/A

Corresponding author(s): Stefan Westermann (stefan.westermann@uni-due.de)

Review Timeline:

Submission Date:	12th Jul 19
Editorial Decision:	14th Aug 19
Resubmission:	20th Mar 20
Editorial Decision:	14th Apr 20
Revision Received:	1st May 20
Accepted:	12th May 20

Editor: Hartmut Vodermaier

Transaction Report:

Thank you for submitting your manuscript on differential Mif2 and Ame1 binding to Mtw1 for our editorial consideration. We have now received the enclosed reports of three expert reviewers, in light of which we unfortunately had to conclude that the study is not a sufficiently strong candidate for EMBO Journal publication, at least not at the present stage. As you will see, the referees acknowledge the rationale of the study and also appreciate the overall quality of the biochemical and genetic analyses. At the same time, they however remain unconvinced that the presented data provide sufficiently conclusive new insights and definitive support for the key conclusions of the study. In particular, all reviewers note the somewhat equivocal/discrepant results on Mif2/Ame1 N-terminal domain roles in vitro and in vivo, which impede straightforward interpretations. They furthermore find that the current evidence for distinct Mif2/Ame1 conformations and Mif2 conformational changes involved in reversible auto-inhibition is insufficient, requiring additional support from complementary biochemical/biophysical/structural approaches. Without repeating all their specific comments in more detail here, I am afraid to say that I therefore find the study at present still too preliminary to warrant concrete further consideration for an EMBO Journal article. That said, should future work allow you to extend the work and obtain more definitive evidence through complementary methods, I would remain open to discussing a potential resubmission of the study at a later point. At the current stage, however, I am unfortunately not able to come to a more encouraging decision on this occasion.

In any case, thank you once more for the opportunity to consider this work, and I hope that you will nevertheless find the detailed comments and suggestions from our referees helpful.

With best regards,

Hartmut

Hartmut Vodermaier, PhD
Senior Editor / The EMBO Journal
h.vodermaier@embojournal.org

Referee #1:

In this manuscript, Killinger et al., aim to explore how kinetochore assembly is limited to the centromere (i.e., does not occur in the nucleoplasm). To do this, they focus on the assembly of the *S. cerevisiae* kinetochore downstream of Mif2 and Ame1 through their ability to directly bind and recruit Mtw1c, the primary organizer of outer kinetochore assembly. The authors define the molecular requirements for Mif2 or Ame1 binding to Mtw1c. They find that the Mif2 N-terminal sequence is sufficient to rescue the function of the Ame1 N-terminus. In contrast, when the N-terminus of Mif2 is swapped for that of Ame1, this mutant is not able to functionally rescue the loss of the Mif2 N-terminus, binds more readily to Mtw1c, and results in chromosome mis-segregation errors and Mtw1 mislocalization in cells. Together, this data leads the authors to propose a model in which Mif2 undergoes a conformational change that prevents its ability to bind the Mtw1c therefore limiting the interaction to occur unless Mif2 is at the centromere.

Overall, the model proposed by the authors is interesting, but is not directly supported by the data presented. There are also many experiments where a modest effect is interpreted in a very strong way. Although there are some important observations here, this manuscript does not meet the standard for EMBOJ in its present form.

1. The interpretation of the mutant data in Figure 2 has several issues. First, the authors present these mutants as being specific to disrupting one interaction (either Ame1 or Mif2), but this is clearly not true based on the functional data. The 5A mutant is predicted to disrupt the non-essential interaction with Mif2, but this mutant is lethal. In contrast, the 3A mutant is predicted to disrupt the interaction with essential Ame1 N-terminal domain, but is viable. The authors also show in vitro that the 3A mutant is no longer able to bind Mtw1C so it is not clear why this mutant would not recapitulate the loss of Ame1N-term. I also note that the essentiality reported for the Mtw1-3A mutant is different in Figure 2B and Figure 2C, and the authors do not provide a compelling explanation for this. This section needs to be set up and interpreted differently to accurately present the results and not force them into a specific model.
2. The authors conclude that Ame1 and Mif2 bind to separate Mtw1c molecules. However, the only data in this paper that directly test this is the competition assay performed Figure 3, which is not fully convincing. The effect of increasing Ame1 concentrations appears quite subtle, with only a modest competition observed at 100 μ M. Quantification of this data would be helpful, but it seems like a minor effect for the strength of the point that they are making. Similarly, addition of soluble Ame1 N terminal peptide does not appear to compete strongly when Mtw1 is pulled down with the Ame1 N terminus. If they want to make a strong point for the competitive nature of the binding, a more rigorous biochemical strategy would be critical here.
3. The authors propose that Mif2 binding to Cse4 relieves an autoinhibition to allow to bind to Mtw1c. This is based in part on the binding assays in Figure 6. However, I don't find these data fully compelling. The shift that the authors observe when combining Cse4n with Mif2 and Ame1 is quite subtle. In addition, there are several possible interpretations beyond a conformational change in Mif2. First, the presence of Cse4n may solely help to increase the local concentration of Mif2 such that it can more readily bind Mtw1. Second, it is possible that they are getting to separate complexes (Mif2-Cse4 and Mif2-Mtw1) and that the presence of both distinct complexes makes it appear that there is a larger shift. As this is a central part of their current model, it would be important to conduct an alternative test to confirm whether the presence of Cse4n actually confers a change in direct affinity.
4. As support for the auto-inhibitory model, the authors reanalyze prior crosslinking data suggesting

that there are contacts within Mif2. However, these experiments don't directly test whether there are conformational changes that require N-terminal exposure. One way to address this would be to assess whether there are differences between the intramolecular crosslinking tracks of wild-type Mif2 and Mif2Ame1-Nterm proteins. Based on their model, there should be fewer of these crosslinks (or none) for the N terminal swapped Mif2.

5. A core finding of this paper is that the related sequences in Ame1 and Mif2 are not functionally identical. This is a nice observation, but there are a variety of explanations for this beyond the idea that Mif2 undergoes a conformational change. Given their ability to conduct these replacement assays and probe the sequence requirements, one way to strengthen this paper would be to generate point mutants within these sequences to test the consequences. For example, they could make specific swap mutations (the Mif2 sequence has a slight increase in positive charges). Even more interesting, it looks like the Mif2 sequence, but not the Ame1 sequence, has the ability to be phosphorylated within the RxxG motif. Post-translational control of binding would be a much more likely explanation for the differential behaviors that they observe in cells, and they should definitely test this model.

6. In Fig 2E, the second Superose spectra is missing the 5A denoting the Mtw1c.

7. In figure 4C, the western to show expression levels of the Ame1 constructs does not agree with the predicted sizes for these. Perhaps the authors inverted some of the labels?

Referee #2:

The kinetochore is a structure containing multi-protein complexes, which connects chromatin with spindle microtubules to ensure accurate chromosome segregation. The kinetochore components are relatively conserved from yeasts to human, and yeasts were widely used as a model system in the kinetochore research field. Using budding yeast kinetochore components, a kinetochore structure has been reconstituted in vitro and its structure is also determined at atomic level, based on Cryo-EM technology. While these in vitro studies are very useful, as the kinetochore is dynamic, assembly mechanisms and their regulation are still largely unknown. The authors in this paper have been studying multiple kinetochore components for a long time using budding yeast as a model system and have done several pioneer works. In this study, they characterized two components Mif2 (CENP-C) and Ame1 (CENP-U), both of which function as a receptor for an outer kinetochore complex Mtw1C (Mis12C). They found that the Mif2 N-terminal region fused with Ame1 suppressed growth defects in cells expressing Ame1 lacking its N-terminal region, but the Ame1 N-terminal region fused with Mif2 did not suppress Mif2 defects of cells expressing Mif2 lacking its N-terminal region (in Δ Ctf19 background). Combined with additional biochemical analyses, the authors proposed that "Mif2 auto-inhibition" is critical for explaining this phenotype. The authors also demonstrate that "Mif2 auto-inhibition" might be cancelled upon binding of Mif2 to Cse4 (CENP-A) nucleosome.

Overall, this paper contains a solid genetics study and biochemical analyses. In addition, "Mif2 auto-inhibition" is an interesting and new concept. Therefore, this would be acceptable for publication in EMBO J, but to conclude on "Mif2 auto-inhibition" they need additional evidence. The authors should address several specific concerns before publication.

1. "Mif2 auto-inhibition" is an interesting idea. But there might be other ways to explain the yeast phenotype they observed. The authors should consider other models and should explain why "Mif2 auto-inhibition" is the most likely model.
2. If "Mif2 auto-inhibition" is correct, the Mtw1 binding region of Mif2 should show similar binding

affinity to that of Ame1 in the absence of N-terminal regions of proteins. They should demonstrate such with biochemical data using Mtw1 binding peptides for Mif2 and Ame1.

3. If "Mif2 auto-inhibition" is an important regulatory mechanism for kinetochore assembly, why is the Mif2 N-terminal region but not the Ame1 N-terminal region dispensable? While I agree that the Ame1N-Mif2 chimera is toxic, I feel that Ame1 is much more important for recruiting Mtw1 to the kinetochore. Please clarify this in Discussion.

4. Although the biochemical experiments in Figure 6 were done well, these do not directly support the authors' idea that Mif2 forms a self-contact structure, while Ame1 forms a "rod-like" structure (Figure 8). If they can use an alternative way (such as AFM imaging?) to directly show these conformations to support their model, it would be good.

Minor issues

1. Figure should be presented in a sequential order. On page6, Fig2D should not be presented before Fig2C. Orders of Fig6 should be organized in a sequential order.

2. The authors sometimes mixed yeast proteins and human proteins. Many proteins have similar roles for yeast proteins and human homologues. But some proteins have specific roles. For example, human CENP-U (Ame1 homologue) does not clearly bind to the Mis12 complex. So, they should not simply apply knowledge from human studies to yeast studies. Yeast Mif2 is not the same as human CENP-C.

3. On page 16, concerning tethering experiment of CENP-C N-terminus, authors cited Gascoigne et al. 2011. This paper does not strictly show induction of functional kinetochores, as Hori et al. (2013, JCB) firstly showed induction of functional kinetochore by the CENP-C N-terminus. In addition to Gascoigne et al. 2011, they should also cite Hori et al. 2013, (JCB).

Referee #3:

Summary and general comments

This study describes a combination of yeast genetic and biochemical experiments investigating how two components of the 'inner' kinetochore of budding yeast, Mif2 and the Ame1-Okp1 complex (AO), interact with the Mtw1 complex (Mtw1c), a key component of the 'outer' kinetochore. This is an important and relatively unexplored area of kinetochore biochemistry, so the topic will be of considerable interest to the field. The primary conclusion drawn by the authors is that Mif2 is intrinsically attenuated in its binding to Mtw1c, and that this auto-inhibition is relieved when Mif2 binds the centromere-specific nucleosome, Cse4n. While the data are consistent with this interpretation, the evidence for auto-inhibition does not seem entirely convincing. More specifically, the evidence that AO competitively inhibits the binding of Mif2 to Mtw1c seems weak. Also, the evidence that a chimeric Mif2, with its normal N-terminus replaced by the corresponding N-terminus from Ame1, binds better to Mtw1c than wild-type Mif2 seems inconsistent - with one experiment showing a large difference but another showing only a very subtle difference, at best. The possibility of auto-inhibition within Mif2 that might be relieved by binding to Cse4n is nevertheless interesting, and potentially very important. The paper includes additional interesting data, such as the assembly in vitro of a large supercomplex including Cse4n, Mif2, AO, and Mtw1, and the demonstration that swapping N-termini between Mif2 and Ame1 in vivo has different impacts on cell viability and chromosome segregation. If the authors can provide more convincing evidence of the auto-inhibition, I would enthusiastically support publication.

Detailed comments

Several mutants of Mtw1c were designed, based on the crystal structure of Dimitrova 2016, to attempt to specifically disrupt the interaction of Mtw1c with Mif2 (Mtw1-5A), or with AO (Mtw1-3A), or with both Mif2 and AO (Mtw1-8A). Because previous work indicated that the N-terminal motif of Mif2 that binds Mtw1c is dispensable, it was expected that the Mtw1-5A mutant might support cell viability, whereas the -3A and -8A mutants should not, given that interaction of Mtw1c with AO is likely essential. However, the -5A mutation unexpectedly interfered with AO binding and did not support viability. One possible explanation is that Mif2 and AO might bind to largely overlapping sites on Mtw1c and therefore might compete against one another for binding to Mtw1c. An attempt was made to demonstrate such competitive inhibition using small peptide fragments of Ame1 (1-20) and Mif2 (1-41) to pull down a heterodimeric subcomplex of Mtw1c (Mtw1-Nnf1). However, these data are not especially convincing because adding up to 100 μ M free Ame1 (1-20) resulted in only a slight decrease in the amount of Mtw1-Nnf1 that pulled down with Mif2 (1-41). Single point mutations in Ame1 (R10A) and Mif2 (R11A) were also designed, based on sequence alignments of proteins from several yeast species, which weaken the interactions with the Mtw1-Nnf1 subcomplex. Altogether these results narrow the relevant residues underlying the interactions between AO, Mif2, and Mtw1c. They suggest but do not prove that Mif2 and AO bind Mtw1c in a competitive, mutually exclusive manner.

A chimeric version of Ame1, in which the N-terminal Mtw1c-binding portion of Ame1 was replaced with that of Mif2, supported cell viability with apparently normal levels of Mtw1-GFP at kinetochores and unperturbed cell cycle progression. This shows that both N-termini have sufficient affinity for Mtw1c to support its essential binding to AO.

By contrast, a chimeric version of Mif2, in which its N-terminus was replaced with that of Ame1, did not function like normal wild-type Mif2. Because removal of the N-terminus of Mif2 is tolerated in a normal strain background, this comparison had to be done in a sensitized *ctf19*-deletion background, in which the N-terminus of Mif2 becomes essential for cell viability. In this background, the chimeric Mif2 with an N-terminus taken from Ame1 did not support viability, which indicates that the Ame1 N-terminus is not equivalent to the Mif2 N-terminus when grafted onto the Mif2 protein. However, it cannot be excluded that a larger grafted portion of Ame1, or a graft onto a slightly different location within the Mif2 sequence, might suffice. Perhaps the specific choice of break-fusion locations, by bad luck, was simply disruptive. The Mif2 chimera appears to be more susceptible to proteolytic degradation during purification from Sf9 cells, which could mean that the grafted protein was, unfortunately, biochemically 'unhappy'. Cross-linking mass spectrometry analysis suggested that the N-terminus of Mif2 might sometimes contact other, more C-terminal portions of the protein, which is not at all surprising for a region that is thought to be mostly disordered. All these behaviors of the Mif2 chimera would be unremarkable without the additional observation that this protein appears to bind better than wild-type Mif2 to Mtw1c. This interpretation is based on immuno-precipitations from Sf9 cells coinfecting with Mif2 and Mtw1c, and on one SEC experiment in which the addition of bacterially expressed Mtw1c induced a large shift of the chimeric Mif2, but only a smaller shift of the wild-type Mif2. It is suggested that this higher affinity extends to two other Mif2-binding partners, Ctf3c-Cnn1-Wip1 and COMA. However, the data are not entirely convincing on this point, because the levels of Mif2 'bait' are inconsistent across control (wild-type Mif2) and test (chimeric Mif2) lanes. More material might have been pulled down in the latter cases simply because more bait was provided.

A series of SEC experiments shows that the Mif2 protein assembles into supercomplexes together with Mtw1c and Cse4n (Cse4-containing nucleosomes). It is suggested that the chimeric Mif2 again

binds more readily than wild-type Mif2 to Mtw1c in these experiments. However, to my eye the results with wild-type Mif2 look very similar here to those with the chimera (comparing Fig 6B versus 6A). Both versions of Mif2 clearly do assemble into large supercomplexes containing Cse4n and Mtw1c. The author's overall interpretation is that binding of wild-type Mif2 to Cse4n relieves an autoinhibition, absent from the chimeric Mif2, that normally reduces its affinity for Mtw1c. However, the hypothesized autoinhibition seems to cause only a very subtle difference in Fig 6B versus 6A.

Overexpression in otherwise wild-type cells of the chimeric Mif2, but not the wild-type or N-terminally truncated versions of Mif2, caused cell death, apparently due to problems with the arrangement of kinetochores during cell division, and severe defects in chromosome segregation. These observations show that the chimeric Mif2 has a dominant, disruptive effect on mitosis in cells.

*** As a service to authors, The EMBO Journal offers the possibility to directly transfer declined manuscripts to another EMBO Press title (EMBO Reports, EMBO Molecular Medicine, Molecular Systems Biology) or to the open access journal Life Science Alliance launched in partnership between EMBO Press, Rockefeller University Press and Cold Spring Harbor Laboratory Press. The full manuscript (including reviewer comments, where applicable and if chosen) will be automatically forwarded to the receiving journal, to allow for fast handling and a prompt decision on your manuscript. For more details of this service, and to transfer your manuscript to another EMBO title please follow this link:

Link Not Available

Many thanks for considering our study “**Auto-Inhibition of Mif2/CENP-C ensures centromere-dependent kinetochore assembly in budding yeast**” (EMBOJ-2019-102938) by Killinger and colleagues as an article in the EMBO J last year. While there was considerable interest in our work by you and by the reviewers, the conclusion at the time was that the work was in a too preliminary stage to be acceptable and that complementary biochemical methodology was required to strengthen the main conclusions of the study. Over the last months, we have made considerable strides to strengthen the study, in particular using the suggested complementary biophysical and biochemical methods to support the main conclusions of the paper. Specifically, we have added the following novel experimental approaches and findings:

1. We have included **Fluorescence Anisotropy** (fluorescence polarization) measurements to directly and quantitatively test the competition between N-terminal Mif2 and Ame1 peptides for binding to the Mtw1 complex (**new Figure 2B**). The experiments unequivocally show that these two receptors target the same binding interface on the Mtw1 complex, a key element for the main conclusions of the paper and the model.
2. We have now used **Isothermal Titration Calorimetry (ITC)** to quantitatively compare the affinity of the isolated N-terminal Ame1 and Mif2 peptides for the Mtw1 complex. This analysis shows that in isolation, the Mif2 N-terminal peptide displays a higher affinity for the Mtw1 complex compared to the Ame1 peptide (**new Figure 5C**). This is a key finding for the auto-inhibition model, as exchange of the Mif2 peptide for the Ame1 peptide in the context of the full-length Mif2 molecule creates a much higher affinity for the Mtw1 complex. This can only be explained if the Mif2 peptide is attenuated in its binding affinity in the context of the full molecule.
3. We have now included **Comparative cross-linking mass spectrometry (XL-MS)** employing short cross-linkers with higher specificity, to compare the topology and conformation of Mif2 in the absence and in the presence of the centromeric nucleosome. We show that the majority of non-redundant intra-Mif2 cross-links between the N-terminal peptide and the centromere-binding regions, are abolished in the presence of the nucleosome, strongly supporting the idea that Mif2 is present in distinct conformations when alone or when bound to the nucleosome (**new Figure 5D**).
4. Based on the comparative cross-links, we have created a **novel Mif2 mutant that lacks the auto-inhibitory domain** that engages with the N-terminal peptide (**new Figure 6 and 8G**). As predicted from the auto-inhibition model, this Mif2 mutant now binds the Mtw1 complex with higher affinity than wild-type Mif2, even though the sequence of the N-terminal peptide is not altered in this situation (**new Figure 6B and 6C**). Crucially, this mutant is toxic for yeast upon overexpression in vivo (**new Figure 8G**).

Taken together, these experiments substantially strengthen our conclusions regarding the regulation of Mif2 via auto-inhibition. They reveal an important property of the Mif2/CENP-C molecule that will be highly relevant for future studies that further try to decipher the molecular mechanisms of kinetochore assembly.

In addition to these experiments, we have addressed all other concerns of the reviewers. We have also re-organized the manuscript and the presentation of the data, to make them more accessible and avoid some confusion that has led to some misunderstandings in the previous version. Please find below your original decision letter a detailed point-by-point rebuttal (**our answers in red print**).

4th Aug 2019

Re: EMBOJ-2019-102938

Auto-Inhibition of Mif2/CENP-C ensures centromere-dependent kinetochore assembly in budding yeast

Dear Stefan,

Thank you for submitting your manuscript on differential Mif2 and Ame1 binding to Mtw1 for our editorial consideration. We have now received the enclosed reports of three expert reviewers, in light of which we unfortunately had to conclude that the study is not a sufficiently strong candidate for EMBO Journal publication, at least not at the present stage. As you will see, the referees acknowledge the rationale of the study and also appreciate the overall quality of the biochemical and genetic analyses. At the same time, they however remain unconvinced that the presented data provide sufficiently conclusive new insights and definitive support for the key conclusions of the study. In particular, all reviewers note the somewhat equivocal/discrepant results on Mif2/Ame1 N-terminal domain roles in vitro and in vivo, which impede straightforward interpretations. They furthermore find that the current evidence for distinct Mif2/Ame1 conformations and Mif2 conformational changes involved in reversible auto-inhibition is insufficient, requiring additional support from complementary biochemical/biophysical/structural approaches. Without repeating all their specific comments in more detail here, I am afraid to say that I therefore find the study at present still too preliminary to warrant concrete further consideration for an EMBO Journal article. That said, should future work allow you to extend the work and obtain more definitive evidence through complementary methods, I would remain open to discussing a potential resubmission of the study at a later point. At the current stage, however, I am unfortunately not able to come to a more encouraging decision on this occasion.

In any case, thank you once more for the opportunity to consider this work, and I hope that you will nevertheless find the detailed comments and suggestions from our referees helpful.

With best regards,

Hartmut

Hartmut Vodermaier, PhD
Senior Editor / The EMBO Journal
h.vodermaier@embojournal.org

Referee #1:

In this manuscript, Killinger et al., aim to explore how kinetochore assembly is limited to the centromere (i.e., does not occur in the nucleoplasm). To do this, they focus on the assembly of the *S. cerevisiae* kinetochore downstream of Mif2 and Ame1 through their ability to directly bind and recruit Mtw1c, the primary organizer of outer kinetochore assembly. The authors define the molecular requirements for Mif2 or Ame1 binding to Mtw1c. They find that the Mif2 N-terminal sequence is sufficient to rescue the function of the Ame1 N-terminus. In contrast, when the N-terminus of Mif2 is swapped for that of Ame1, this mutant is not able to functionally rescue the loss of the Mif2 N-terminus, binds more readily to Mtw1c, and results in chromosome mis-segregation errors and Mtw1 mislocalization in cells. Together, this data leads the authors to propose a model in which Mif2 undergoes a conformational change that prevents its ability to bind the Mtw1c therefore limiting the interaction to occur unless Mif2 is at the centromere.

Overall, the model proposed by the authors is interesting, but is not directly supported by the data presented. There are also many experiments where a modest effect is interpreted in a very strong way. Although there are some important observations here, this manuscript does not meet the standard for EMBOJ in its present form.

1. The interpretation of the mutant data in Figure 2 has several issues. First, the authors present these mutants as being specific to disrupting one interaction (either Ame1 or Mif2), but this is clearly not true based on the functional data.

This is a misunderstanding that has caused some confusion. The prediction of two separate binding interfaces for Ame1 and Mif2 on Mtw1c stems from the study by Dimitrova et al 2016, in which they interpret structural and biochemical data in this way (see for example their model in Dimitrova et al., 2016, Figure 6D). We have solely used their interpretation as a starting point for our experiments and come to a different conclusion. However, as this is obviously confusing, we have rearranged the Figures and described the experiments in a different way: We now put forward two different possibilities for Mif2/Ame1-Mtw1c configurations (scenarios 1 and 2) and then start with more straightforward biochemical experiments that test the idea of competition between Ame1 and Mif2 for Mtw1c binding directly.

The 5A mutant is predicted to disrupt the non-essential interaction with Mif2, but this mutant is lethal. In contrast, the 3A mutant is predicted to disrupt the interaction with essential Ame1 N-terminal domain, but is viable. The authors also show in vitro that the 3A mutant is no longer able to bind Mtw1C so it is not clear why this mutant would not recapitulate the loss of Ame1N-term.

As stated above, our experiment falsifies the hypothesis (based on the available crystal structure), that the Mtw1-5A mutation would only disrupt Mif2-N binding. We rewrote the text so that it is clear that we interpret the 5A mutant to disrupt both the Ame1 and Mif2 interface. The 3A mutant probably retains some viability after sporulation and dissection, because it only disrupts a part of the Ame1-binding interface, consistent with the finding that interface I (mutated in the 5A version) also makes a critical contribution to Ame1 binding. These experiments support our idea of Ame1 and Mif2 both binding to the same interface.

I also note that the essentiality reported for the Mtw1-3A mutant is different in Figure 2B and Figure 2C, and the authors do not provide a compelling explanation for this. This section needs to be set up and interpreted differently to accurately present the results and not force them into a specific model.

We agree, that the result for the 3A mutant is different after tetrad dissection versus in the anchor-away system. In both cases, however, we obtain either lethality or slow growth, showing that this mutant clearly is compromised. The strain for the anchor-away system contains a number of additional mutations in the background (*tor1-1*, *fpr1* deletion, tagged anchor protein, etc.) which may further compromise growth of the 3A mutant. We have added an explanatory sentence to the respective section.

2. The authors conclude that Ame1 and Mif2 bind to separate Mtw1c molecules. However, the only data in this paper that directly test this is the competition assay performed Figure 3, which is not fully convincing. The effect of increasing Ame1 concentrations appears quite subtle, with only a modest competition observed at 100 μ M. Quantification of this data would be helpful, but it seems like a minor effect for the strength of the point that they are making. Similarly, addition of soluble Ame1 N terminal peptide does not appear to compete strongly when Mtw1 is pulled down with the Ame1 N terminus. If they want to make a strong point for the competitive nature of the binding, a more rigorous biochemical strategy would be critical here.

We agree with the reviewer, that the effect seen in the solid phase binding assay is not very strong. As the reviewer pointed out, Ame1 competes rather weakly with itself, pointing to a limitation of the experimental setup. We took up the suggestion of a more rigorous biochemical strategy and now performed fluorescence polarization assays in which we first determined the affinity of Mif2-N binding to

MNc by using a FITC labeled Mif2 1-41 peptide (**new Figure 2B**). From this measurement we determined the EC50 value and used constant concentrations of FITC-Mif2¹⁻⁴¹ and MNc, which we now competed with unlabeled Ame1¹⁻²⁰ peptide. The results show that the Ame1 peptide competes very effectively in a dose-dependent manner with Mif2-MN binding, yielding a dissociation constant for Ame1 in the low micromolar range.

3. The authors propose that Mif2 binding to Cse4 relieves an autoinhibition to allow to bind to Mtw1c. This is based in part on the binding assays in Figure 6. However, I don't find these data fully compelling. The shift that the authors observe when combining Cse4n with Mif2 and Ame1 is quite subtle. In addition, there are several possible interpretations beyond a conformational change in Mif2. First, the presence of Cse4n may solely help to increase the local concentration of Mif2 such that it can more readily bind Mtw1. Second, it is possible that they are getting to separate complexes (Mif2-Cse4 and Mif2-Mtw1) and that the presence of both distinct complexes makes it appear that there is a larger shift. As this is a central part of their current model, it would be important to conduct an alternative test to confirm whether the presence of Cse4n actually confers a change in direct affinity.

As amounts of full-length Mif2- wt or swap proteins are limiting and therefore assays to quantify binding affinities are very challenging, we decided to 1. Repeat SEC runs under more stringent conditions, 2. Perform comparative x-link and MS analysis of Mif2wt and Mif2wt bound to Cse4n and 3. Construct additional Mif2 mutants that test the auto-inhibition model, in particular those in which the original Mif2 N-terminus remains unaltered.

1. By conducting the SEC experiments under slightly more stringent conditions (increase in ionic strength of the buffer) the shift of Mif2 bound to Mtw1c is now more noticeably different than of Mif2 to Mtw1c in the presence of Cse4n. In addition, Mif2wt plus Cse4n and Mif2swap without Cse4n shift very similarly when combined with Mtw1c (**new Figure 7**). These new conditions make it also easier to appreciate the difference between Mif2 wt and Mif2 swap in the binding to Mtw1c, which has been obvious in Figure 5B (and Supplementary Figure 3), but was more subtle in the previous SEC experiments with the nucleosome.
2. Rather than resorting to already published crosslinking data we now tried to detect conformational differences between Mif2 and Mif2 bound to Cse4 directly. We isolated the samples by gel filtration, used a rather short crosslinker (glutarate-based, 7.7 Å linker) and crosslinked for only a short time (5 minutes) to be able to detect differences between the two samples. Indeed, intramolecular crosslinks of Mif2 that connect the N-terminus to parts of Mif2 involved in Cse4n binding are largely lost in the presence of Cse4n (**new Figure 5D**). This is a direct and clear indication that Mif2 undergoes a conformational change. We presented the data in a way that points out the differences between both samples by presenting all crosslinks that can be only found in one of the two samples highlighted in a different color (blue).
3. Based on the crosslinking results we constructed a Mif2 mutant that retains the original N-terminus but lacks signature motif and AT hook. The auto-inhibition model predicts that without these sequences the N-terminus should be more readily available for binding to Mtw1c. Indeed, **new Figure 6** shows that Mif2 deltaSigAT is a better MNc binder than Mif2-wild type. Importantly, overexpression of this mutant in yeast cells phenocopies the effect of overexpressing Mif2 swap: It compromises cell growth in a manner that depends on the ability to bind the Mtw1c (**Figure 8G**). This shows that different mutants that compromise the auto-inhibited conformation of Mif2 have the same effect in vivo.

4. As support for the auto-inhibitory model, the authors reanalyze prior crosslinking data suggesting that there are contacts within Mif2. However, these experiments don't directly test whether there are conformational changes that require N-terminal exposure. One way to address this would be to assess whether there are differences between the intramolecular crosslinking tracks of wild-type Mif2 and

Mif2Ame1-Nterm proteins. Based on their model, there should be fewer of these crosslinks (or none) for the N terminal swapped Mif2.

As in the swap mutant the N-terminal sequence is different, we reasoned it would be difficult to compare Mif2-wt versus Mif2 swap, simply because different Lysines would be available. We therefore rather favored the strategy of comparing of Mif2 with and without Cse4n (see point 3 above).

5. A core finding of this paper is that the related sequences in Ame1 and Mif2 are not functionally identical. This is a nice observation, but there are a variety of explanations for this beyond the idea that Mif2 undergoes a conformational change. Given their ability to conduct these replacement assays and probe the sequence requirements, one way to strengthen this paper would be to generate point mutants within these sequences to test the consequences. For example, they could make specific swap mutations (the Mif2 sequence has a slight increase in positive charges).

We introduced the point mutation R10E into the Mif2 swap protein. A prediction from our findings is that overexpression of Mif2^{Ame1R10E} should phenocopy Mif2deltaN and revert the toxicity of Mif2 swap in the overexpression setting, because this mutant cannot bind Mtw1c (**new Figure 8B**). We can clearly see this result in the serial dilution assay. Apart from this, we reasoned it would be difficult to interpret other point mutants in the N-terminal domain, because they could either influence the affinity for Mtw1c directly, or indirectly via influencing the protein conformation. We therefore rather constructed the Mif2deltaSigAT mutant, in which the original N-terminus remains unchanged, but regions involved in the inhibition of the N-terminus are eliminated. The analysis of this mutant is presented in the new **Figure 6** and **Figure 8G**.

Even more interesting, it looks like the Mif2 sequence, but not the Ame1 sequence, has the ability to be phosphorylated within the RxxG motif. Post-translational control of binding would be a much more likely explanation for the differential behaviors that they observe in cells, and they should definitely test this model.

We thank the reviewer for these suggestions. We considered the possibility that the purified wild-type Mif2 may be inhibited in binding to Mtw1c via phosphorylation of the Mif2-specific N-terminus (this may well occur in the Sf9 cells), while the Mif2 swap mutant is not. We therefore conducted SEC experiments with Mif2 and Mtw1c with and without lambda phosphatase treatment of Mif2 (**see Figure R1 below**). We did not find that lambda phosphatase treatment improves binding of Mif2-wt to Mtw1c in our in vitro experiments. We note that also the Ame1 N-terminus contains phosphorylatable residues. It seems likely that phosphorylation can play a role in the regulation of Ame1/Mif2 binding to Mtw1c, this is clearly the case for Dsn1 phosphorylation by Ipl1. In this study, however, we chose to focus on intrinsic properties of the Mif2 molecule that are revealed when the N-terminus is exchanged.

Figure R1

6. In Fig 2E, the second Superose spectra is missing the 5A denoting the Mtw1c.

Thanks, we added the missing description.

7. In figure 4C, the western to show expression levels of the Ame1 constructs does not agree with the predicted sizes for these. Perhaps the authors inverted some of the labels?

The Ame1 protein with a 3xHA tag has a molecular weight of about 41 kDa. The bands run slightly higher, but are definitely in the predicted range. Higher molecular weight bands probably represent different phosphorylated forms of Ame1, which is something that we often see when working with Ame1. Ame1 delta N appears to migrate slightly slower than Ame1-wt, but this may be due to a different phosphoform.

Referee #2:

The kinetochore is a structure containing multi-protein complexes, which connects chromatin with spindle microtubules to ensure accurate chromosome segregation. The kinetochore components are relatively conserved from yeasts to human, and yeasts were widely used as a model system in the kinetochore research field. Using budding yeast kinetochore components, a kinetochore structure has been reconstituted in vitro and its structure is also determined at atomic level, based on Cryo-EM technology. While these in vitro studies are very useful, as the kinetochore is dynamic, assembly mechanisms and their regulation are still largely unknown. The authors in this paper have been studying multiple kinetochore components for a long time using budding yeast as a model system and have done several pioneer works. In this study, they characterized two components Mif2 (CENP-C) and Ame1 (CENP-U), both of which function as a receptor for an outer kinetochore complex Mtw1C (Mis12C). They found that the Mif2 N-terminal region fused with Ame1 suppressed growth defects in cells expressing Ame1 lacking its N-terminal region, but the Ame1 N-terminal region fused with Mif2 did not suppress Mif2 defects of cells expressing Mif2 lacking its N-terminal region (in Δ Ctf19 background). Combined with additional biochemical analyses, the authors proposed that "Mif2 auto-inhibition" is critical for explaining this phenotype. The authors also demonstrate that "Mif2 auto-inhibition" might be cancelled upon binding of Mif2 to Cse4 (CENP-A) nucleosome.

Overall, this paper contains a solid genetics study and biochemical analyses. In addition, "Mif2 auto-inhibition" is an interesting and new concept. Therefore, this would be acceptable for publication in EMBO J, but to conclude on "Mif2 auto-inhibition" they need additional evidence. The authors should address several specific concerns before publication.

1. "Mif2 auto-inhibition" is an interesting idea. But there might be other ways to explain the yeast phenotype they observed. The authors should consider other models and should explain why "Mif2 auto-inhibition" is the most likely model.

Please see our response to reviewer 1, points 3 and 5, covering these points.

2. If "Mif2 auto-inhibition" is correct, the Mtw1 binding region of Mif2 should show similar binding affinity to that of Ame1 in the absence of N-terminal regions of proteins. They should demonstrate such with biochemical data using Mtw1 binding peptides for Mif2 and Ame1.

We agree with the reviewer that this is an important point and have therefore conducted ITC experiments for the isolated Ame1 and Mif2 peptides (**new Figure 5C**). The isolated Mif2 peptide has a higher affinity for MN than the Ame1 peptide, making it even more surprising that full-length Mif2 is such a weak Mtw1c binder, but turned into a strong binder when the weaker Ame1-N peptide is swapped in.

3. If "Mif2 auto-inhibition" is an important regulatory mechanism for kinetochore assembly, why is the Mif2 N-terminal region but not the Ame1 N-terminal region dispensable? While I agree that the Ame1N-Mif2 chimera is toxic, I feel that Ame1 is much more important for recruiting Mtw1 to the kinetochore. Please clarify this in Discussion.

We have amended the discussion to include these points. In a wild-type background loss of the Mif2-N-terminus is tolerated. The "missing" connection to those Mtw1c molecules normally bound to Mif2-N can probably be compensated for by additional contacts within the kinetochore. Therefore, it seems that lack of Mtw1c binding by Mif2 is more readily tolerated than "too much" (inappropriate or unscheduled) binding. In the ctf19 delta background, Mtw1c binding by Mif2 clearly becomes relevant. In a wild-type background, Ame1-N might be more important, simply because normally more Mtw1c molecules might be connected to Ame1 than to Mif2, i.e. the copy number of Ame1 at the kinetochore might exceed that of Mif2.

4. Although the biochemical experiments in Figure 6 were done well, these do not directly support the authors' idea that Mif2 forms a self-contact structure, while Ame1 forms a "rod-like" structure (Figure 8). If they can use an alternative way (such as AFM imaging?) to directly show these conformations to support their model, it would be good.

We have now included comparative crosslinking followed by mass spectrometry to explore the different topologies of Mif2 when in isolation, or when bound to the Cse4 nucleosome (**new Figure 5D**). We also feel that including the new Mif2deltaSigAT mutant, in which the N-terminal binding domain is unchanged, but which displays increased affinity for MNc (**new Figure 6**), strongly supports our conclusions.

Minor issues

1. Figure should be presented in a sequential order. On page6, Fig2D should not be presented before Fig2C. Orders of Fig6 should be organized in a sequential order.

Thanks for the remark, we arranged the figure accordingly.

2. The authors sometimes mixed yeast proteins and human proteins. Many proteins have similar roles for yeast proteins and human homologues. But some proteins have specific roles. For example, human CENP-U (Ame1 homologue) does not clearly bind to the Mis12 complex. So, they should not simply apply knowledge from human studies to yeast studies. Yeast Mif2 is not the same as human CENP-C.

We changed all CENP-U and CENP-C to Ame1 and Mif2, except when it is to state that they are homologs or when citing studies in which CENP-U or CENP-C were used.

3. On page 16, concerning tethering experiment of CENP-C N-terminus, authors cited Gascoigne et al. 2011. This paper does not strictly show induction of functional kinetochores, as Hori et al. (2013, JCB) firstly showed induction of functional kinetochore by the CENP-C N-terminus. In addition to Gascoigne et al. 2011, they should also cite Hori et al. 2013, (JCB).

Thanks for the comment, we have added the citation.

Referee #3:

Summary and general comments

This study describes a combination of yeast genetic and biochemical experiments investigating how two components of the 'inner' kinetochore of budding yeast, Mif2 and the Ame1-Okp1 complex (AO), interact with the Mtw1 complex (Mtw1c), a key component of the 'outer' kinetochore. This is an important and relatively unexplored area of kinetochore biochemistry, so the topic will be of considerable interest to the field. The primary conclusion drawn by the authors is that Mif2 is intrinsically attenuated in its binding to Mtw1c, and that this auto-inhibition is relieved when Mif2 binds the centromere-specific nucleosome, Cse4n. While the data are consistent with this interpretation, the evidence for auto-inhibition does not seem entirely convincing. More specifically, the evidence that AO competitively inhibits the binding of Mif2 to Mtw1c seems weak. Also, the evidence that a chimeric Mif2, with its normal N-terminus replaced by the corresponding N-terminus from Ame1, binds better to Mtw1c than wild-type Mif2 seems inconsistent - with one experiment showing a large difference but another showing only a very subtle difference, at best. The possibility of auto-inhibition within Mif2 that might be relieved by binding to Cse4n is nevertheless interesting, and potentially very important. The paper includes additional interesting data, such as the assembly in vitro of a large supercomplex including Cse4n, Mif2, AO, and Mtw1, and the demonstration that swapping N-termini between Mif2 and Ame1 in vivo has different impacts on cell viability and chromosome segregation. If the authors can provide more convincing evidence of the auto-inhibition, I would enthusiastically support publication.

Detailed comments

Several mutants of Mtw1c were designed, based on the crystal structure of Dimitrova 2016, to attempt to specifically disrupt the interaction of Mtw1c with Mif2 (Mtw1-5A), or with AO (Mtw1-3A), or with both Mif2 and AO (Mtw1-8A). Because previous work indicated that the N-terminal motif of Mif2 that binds Mtw1c is dispensable, it was expected that the Mtw1-5A mutant might support cell viability, whereas the -3A and -8A mutants should not, given that interaction of Mtw1c with AO is likely essential. However, the -5A mutation unexpectedly interfered with AO binding and did not support viability. One possible explanation is that Mif2 and AO might bind to largely overlapping sites on Mtw1c and therefore might compete against one another for binding to Mtw1c. An attempt was made to demonstrate such competitive inhibition using small peptide fragments of Ame1 (1-20) and Mif2 (1-41) to pull down a heterodimeric subcomplex of Mtw1c (Mtw1-Nnf1). However, these data are not especially convincing because adding up to 100 uM free Ame1 (1-20) resulted in only a slight decrease in the amount of Mtw1-Nnf1 that pulled down with Mif2 (1-41). Single point mutations in Ame1 (R10A) and Mif2 (R11A) were also designed, based on sequence alignments of proteins from several yeast species, which weaken the

interactions with the Mtw1-Nnf1 subcomplex. Altogether these results narrow the relevant residues underlying the interactions between AO, Mif2, and Mtw1c. They suggest but do not prove that Mif2 and AO bind Mtw1c in a competitive, mutually exclusive manner.

See comments for Referee 1, point 2: we have now used Fluorescence polarization assays (new Figure 2B) to demonstrate the point of competition more rigorously.

A chimeric version of Ame1, in which the N-terminal Mtw1c-binding portion of Ame1 was replaced with that of Mif2, supported cell viability with apparently normal levels of Mtw1-GFP at kinetochores and unperturbed cell cycle progression. This shows that both N-termini have sufficient affinity for Mtw1c to support its essential binding to AO.

By contrast, a chimeric version of Mif2, in which its N-terminus was replaced with that of Ame1, did not function like normal wild-type Mif2. Because removal of the N-terminus of Mif2 is tolerated in a normal strain background, this comparison had to be done in a sensitized *ctf19*-deletion background, in which the N-terminus of Mif2 becomes essential for cell viability. In this background, the chimeric Mif2 with an N-terminus taken from Ame1 did not support viability, which indicates that the Ame1 N-terminus is not equivalent to the Mif2 N-terminus when grafted onto the Mif2 protein. However, it cannot be excluded that a larger grafted portion of Ame1, or a graft onto a slightly different location within the Mif2 sequence, might suffice. Perhaps the specific choice of break-fusion locations, by bad luck, was simply disruptive. The Mif2 chimera appears to be more susceptible to proteolytic degradation during purification from Sf9 cells, which could mean that the grafted protein was, unfortunately, biochemically 'unhappy'.

We appreciate the concern but find that the Mif2 Ame1-N chimera does not suffer from a lack of expression or lack of Mtw1 binding. In the revised manuscript we have included a co-immunoprecipitation experiment from yeast extracts (new Figure 8F). This experiment, using Mif2 variants expressed under the endogenous promoter, shows that the Mif2 swap is expressed well and, in full agreement with the *in vitro* data, it co-IPs an increased amount of Mtw1-myc from cell extracts. Thus, rather than being non-functional, it appears to bind Mtw1c better.

Cross-linking mass spectrometry analysis suggested that the N-terminus of Mif2 might sometimes contact other, more C-terminal portions of the protein, which is not at all surprising for a region that is thought to be mostly disordered. All these behaviors of the Mif2 chimera would be unremarkable without the additional observation that this protein appears to bind better than wild-type Mif2 to Mtw1c. This interpretation is based on immuno-precipitations from Sf9 cells coinfecting with Mif2 and Mtw1c, and on one SEC experiment in which the addition of bacterially expressed Mtw1c induced a large shift of the chimeric Mif2, but only a smaller shift of the wild-type Mif2. It is suggested that this higher affinity extends to two other Mif2-binding partners, Ctf3c-Cnn1-Wip1 and COMA. However, the data are not entirely convincing on this point, because the levels of Mif2 'bait' are inconsistent across control (wild-type Mif2) and test (chimeric Mif2) lanes. More material might have been pulled down in the latter cases simply because more bait was provided.

Please see our comments to Reviewer 1, point 3, these cover these concerns. Please note that we have also added a co-IP experiment from yeast extracts of cells expressing Mif2Flag and Mtw1Myc. We clearly see that Mif2swap pulls down more Mtw1 than Mif2 wild type (new Figure 8F). In the new manuscript we have removed the Co-IP experiments with Ctf3c-Cnn1-Wip1 and COMA, because they distracted from a more focused analysis of Mif2-Mtw1c binding.

A series of SEC experiments shows that the Mif2 protein assembles into supercomplexes together with

Mtw1c and Cse4n (Cse4-containing nucleosomes). It is suggested that the chimeric Mif2 again binds more readily than wild-type Mif2 to Mtw1c in these experiments. However, to my eye the results with wild-type Mif2 look very similar here to those with the chimera (comparing Fig 6B versus 6A). Both versions of Mif2 clearly do assemble into large supercomplexes containing Cse4n and Mtw1c. The author's overall interpretation is that binding of wild-type Mif2 to Cse4n relieves an autoinhibition, absent from the chimeric Mif2, that normally reduces its affinity for Mtw1c. However, the hypothesized autoinhibition seems to cause only a very subtle difference in Fig 6B versus 6A.

We have now repeated the SEC experiments under more stringent conditions to make the differences more obvious (new Figure 7). It should now be clear that Mif2wt only weakly binds to Mtw1c in solution. By contrast, Mif2swap itself is a good binder of Mtw1c, as it shifts more readily and forms a complex with Mtw1c even in the absence of Cse4n. In the presence of Cse4n both Mif2 variants form supercomplexes with Cse4n and Mtw1c, which is what we would expect. The important difference is that Mif2wt only forms a complex with Mtw1c in the presence of Cse4n, whereas Mif2swap already binds Mtw1c on its own, shifting comparably to Mif2wt Mtw1c Cse4n. Please also note the other experiments described in the response to Reviewer 1, point 3, that touch on additional points regarding auto-inhibition.

Overexpression in otherwise wild-type cells of the chimeric Mif2, but not the wild-type or N-terminally truncated versions of Mif2, caused cell death, apparently due to problems with the arrangement of kinetochores during cell division, and severe defects in chromosome segregation. These observations show that the chimeric Mif2 has a dominant, disruptive effect on mitosis in cells

Please note, that we have included the novel Mif2deltaSigAT mutant, which compromises auto-inhibition of the Mif2 N-terminus in vitro (new Figure 6). We show that overexpression of this mutant, just like overexpression of chimeric Mif2 Ame1-N, is toxic for cells (new Figure 8G), strongly supporting our model.

Thank you for submitting a new version of your manuscript on Mif2 autoinhibition regulation during kinetochore assembly for our consideration. All three original referees have now assessed it again, and agree that the study has been significantly improved and should now in principle be suitable for EMBO Journal publication. At the same time, they still list a number of specific points that would need to be satisfactorily clarified/modified prior to acceptance, and I would herewith like to invite you to address these issues in a final round of minor revision.

When preparing this final version, please make sure to adhere as closely as possible to our author guidelines for revisions and formatting, as this should greatly facilitate all further processing. Some particular editorial points to pay attention to are the following:

Referee #1:

In this manuscript, Killinger et al. explore how kinetochore assembly is limited to the centromere (i.e., does not occur in the nucleoplasm). To do this, they focus on the assembly of the *S. cerevisiae* kinetochore downstream of Mif2 and Ame1 through their ability to directly bind and recruit Mtw1c, the primary organizer of outer kinetochore assembly. The authors define the molecular requirements for Mif2 or Ame1 binding to Mtw1c. They find that the Mif2 N-terminal sequence is sufficient to rescue the function of the Ame1 N-terminus. In contrast, when the N-terminus of Mif2 is swapped for that of Ame1, this mutant is not able to functionally rescue the loss of the Mif2 N-terminus, binds more readily to Mtw1c, and results in chromosome mis-segregation errors and Mtw1c mislocalization in cells. Additionally, they show that expressing a mutant of Mif2, in which the auto-inhibition domain is lost, results in similar behavior to that of swapping the N-terminal sequence. Together, this data leads the authors to propose a model in which Mif2 undergoes a conformational change that prevents its ability to bind the Mtw1c therefore limiting the interaction to occur unless Mif2 is at the centromere.

The new work added by the authors has made significant improvements to the paper. The change in narrative as well as the inclusion of additional experiments more strongly supports the claims and model set up by the authors in this paper. In particular, the incorporation of the Mif2- Δ sigAT mutant as well as the more quantitative biochemical approaches were a nice addition that added convincing evidence to support their model. Based on this, I fully support publication of this paper (after the authors address the minor comments below).

Minor Comments:

1. Figure 2A - it would be nice to include some quantification of the gel to show the reduction in levels of their protein in question. I would apply this comment to figure 5A as well. Also, from looking at the paper, it appears that the gel in this figure is identical to the one in Figure 3 of the old paper, but the concentrations are labeled differently. I am just curious about the basis for this change.
2. In Fig 3E and figure 7, the addition of boxes around the earlier fractions would make it a bit easier to interpret the changes in elution for proteins shown in these gels.
3. In the final paragraph of the results section the authors incorrectly reference figure 8G and 8F, I believe these need to be swapped.

Referee #2:

This manuscript is a revised version, which was submitted to EMBO J a year ago. The previous MS was potentially interesting, but evidences for auto-inhibition of Mif2 were not sufficient for publication. Here, authors performed additional experiments to support with authors' idea and the MS was substantially improved. I appreciate crosslink-mass spec experiments and ITC experiments, which strongly support with authors' idea. Therefore, I recommend publication of this work in EMBO J.

Minor points

1. Figure 1. Although authors described "CCAN (CENP-C)" in general KT plan, I feel that there are CENP-C and CENP-T dual pathways in general plan. It may be better to add CENP-T to CENP-C in general KT plan.
2. Auto-inhibition of Mif2 is an interesting regulation. Why does not Ame1 have this regulation, because Ame1 also has Cse4 binding activity? In addition, if N-terminus of Ame1 binds the MIND complex in interphase, this might be deleterious for cells? Although I know this point is beyond the scope of this MS, it would be good if authors can add their ideas to discussion.

Referee #3:

With the extensive revisions, I feel that this paper is much improved. In particular, stronger evidence is presented in support of the arguments for auto-inhibition within full-length Mif2 and competition between Mif2 and Ame1 for binding to Mtw1c. There are still a few issues, listed below, and places where the authors seem to overstate their case and I would recommend more careful wording. But these overstatements seem relatively minor and, despite the bold language, readers can decide for themselves what they truly believe based on the actual data. The concept that Mif2 is auto-inhibited until it binds the centromere is interesting, especially given the previously described examples of similar auto-inhibition within other kinetochore complexes. (The parallels are nicely described by the authors in their discussion.) Assuming the corrections below are made I am happy to support publication in EMBO J.

Fluorescence polarization experiment, Figure 2B: I think there is either a mislabeling of the axes here, or else an error of analysis (probably the former). The lower graph indicates that about 3 millimolar of Ame1[1-20] was needed for half-maximal reduction of "r". (i.e., about $10^{-2.5}$ molar at half-max, based on the labeling of the axis.) But the table below the graph and the corresponding text on page 6 claims an affinity that is three orders of magnitude stronger (around 2 micromolar).

Competition experiment of Figure 2A: Despite the new fluorescence polarization data, it remains clear that adding as much as 65 micromolar of Ame1[1-20] caused only a very minor drop in the amount of MN that was pulled down here by Mif2-N. Thus the competition between these two N-terminal peptides for binding MN is relatively weak by this measure. Why don't the two measures (pull-down versus fluorescence polarization) agree on this point? Given the inconsistency, it seems overstated to say that these interactions are "proved to be mutually exclusive" (page 5).

Typo, page 7: "conserved Glycin" should be Glycine.

About the Western blot in Figure 3D: "The inability of these alleles to support growth was not due to lack of protein expression, as shown by western blot analysis (Figure 3d)." This seems to be an over-interpretation. The blot shows that levels of all the mutants are clearly reduced compared to

wildtype levels. Thus, contrary to what is claimed here in the text, it seems that the lower levels of all the mutant proteins could indeed explain their inability to support growth.

On page 17, in the last three sentences before Discussion, it seems the references to Figures 8G and 8F are inadvertently swapped.

Page 30, methods for fluorescence polarization are omitted: "...with a X and the software X". Also on subsequent page, what is meant by "preceded by excitation wavelength..."?

we are delighted that you and the reviewers find our study now suitable for publication in the EMBO J. We have addressed the remaining questions and made the requested editorial and textual changes in this re-revised version.

Please find below our point-by point rebuttal to the remaining points (our answers in red print).

Referee #1:

In this manuscript, Killinger et al. explore how kinetochore assembly is limited to the centromere (i.e., does not occur in the nucleoplasm). To do this, they focus on the assembly of the *S. cerevisiae* kinetochore downstream of Mif2 and Ame1 through their ability to directly bind and recruit Mtw1c, the primary organizer of outer kinetochore assembly. The authors define the molecular requirements for Mif2 or Ame1 binding to Mtw1c. They find that the Mif2 N-terminal sequence is sufficient to rescue the function of the Ame1 N-terminus. In contrast, when the N-terminus of Mif2 is swapped for that of Ame1, this mutant is not able to functionally rescue the loss of the Mif2 N-terminus, binds more readily to Mtw1c, and results in chromosome mis-segregation errors and Mtw1 mislocalization in cells. Additionally, they show that expressing a mutant of Mif2, in which the auto-inhibition domain is lost, results in similar behavior to that of swapping the N-terminal sequence. Together, this data leads the authors to propose a model in which Mif2 undergoes a conformational change that prevents its ability to bind the Mtw1c therefore limiting the interaction to occur unless Mif2 is at the centromere.

The new work added by the authors has made significant improvements to the paper. The change in narrative as well as the inclusion of additional experiments more strongly supports the claims and model set up by the authors in this paper. In particular, the incorporation of the Mif2- Δ sigAT mutant as well as the more quantitative biochemical approaches were a nice addition that added convincing evidence to support their model. Based on this, I fully support publication of this paper (after the authors address the minor comments below).

Minor Comments:

1. Figure 2A - it would be nice to include some quantification of the gel to show the reduction in levels of their protein in question. I would apply this comment to figure 5A as well.

We have included a quantification for Figure 2A. The integrated density of the Coomassie-stained bands in the raw images were measured in ImageJ and the background subtracted. For Figure 2A, quotient values of the Mtw1/GST-Mif2 bands in the sample without Ame1 competitor were set to 1 and the other values calculated accordingly. The result is now plotted as a bar graph below the gel. For Figure 5A we think that the result is very clear already and that adding a quantification will not improve the Figure, keeping in mind that quantification of western blots can be problematic. In addition, in the same Figure, the point is made by Figure 5B and Figure EV2, both displaying the increased affinity of Mtw1c for Mif2-Ame1N swap.

Also, from looking at the paper, it appears that the gel in this figure is identical to the one in Figure 3 of the old paper, but the concentrations are labeled differently. I am just curious about the basis for this change.

We realized that the Ame1 peptide was lyophilized in the presence of TFA (Trifluoroacetic acid) and therefore corrected the molar concentration taking into account TFA binding to positively charged amino acids.

2. In Fig 3E and figure 7, the addition of boxes around the earlier fractions would make it a bit easier to interpret the changes in elution for proteins shown in these gels.

Thanks for the suggestion, we added boxes to all gel images and also find it easier to interpret.

3. In the final paragraph of the results section the authors incorrectly reference figure 8G and 8F, I believe these need to be swapped.

Thanks for spotting this mistake, we changed the figure references accordingly.

Referee #2:

This manuscript is a revised version, which was submitted to EMBO J a year ago. The previous MS was potentially interesting, but evidences for auto-inhibition of Mif2 were not sufficient for publication. Here, authors performed additional experiments to support with authors' idea and the MS was substantially improved. I appreciate crosslink-mass spec experiments and ITC experiments, which strongly support with authors' idea. Therefore, I recommend publication of this work in EMBO J.

Minor points

1. Figure 1. Although authors described "CCAN (CENP-C)" in general KT plan, I feel that there are CENP-C and CENP-T dual pathways in general plan. It may be better to add CENP-T to CENP-C in general KT plan.

Although we focus our study on the CENP-C pathway for outer kinetochore assembly, we agree that it is helpful to show a more complete picture in Figure 1. We therefore added CENP-T and Cnn1 to the general assembly schemes of the kinetochore and added a sentence in the introduction.

2. Auto-inhibition of Mif2 is an interesting regulation. Why does not Ame1 have this regulation, because Ame1 also has Cse4 binding activity? In addition, if N-terminus of Ame1 binds the MIND complex in interphase, this might be deleterious for cells? Although I know this point is beyond the scope of this MS, it would be good if authors can add their ideas to discussion.

We have added an additional paragraph to the discussion, speculating about these points.

Referee #3:

With the extensive revisions, I feel that this paper is much improved. In particular, stronger evidence is presented in support of the arguments for auto-inhibition within full-length Mif2 and competition between Mif2 and Ame1 for binding to Mtw1c. There are still a few issues, listed below, and places where the authors seem to overstate their case and I would recommend more careful wording. But these overstatements seem relatively minor and, despite the bold language, readers can decide for themselves what they truly believe based on the actual data. The concept that Mif2 is auto-inhibited until it binds the centromere is interesting, especially given the previously described examples of similar auto-inhibition within other kinetochore complexes. (The parallels are nicely described by the authors in their discussion.) Assuming the corrections below are made I am happy to support publication in EMBO J.

Fluorescence polarization experiment, Figure 2B: I think there is either a mislabeling of the axes here, or else an error of analysis (probably the former). The lower graph indicates that about 3 millimolar of Ame1[1-20] was needed for half-maximal reduction of "r". (I.e., about $10^{-2.5}$ molar at half-max, based on the labeling of the axis.) But the table below the graph and the corresponding text on page 6 claims an affinity that is three orders of magnitude stronger (around 2 micromolar).

Thanks for spotting this. We have corrected the labeling of the x-axis in Figure 2B lower graph. We have also included more details regarding the calculation of K_i from the competition experiment in Materials and Methods.

For your information here again a plot of the raw data from the competition experiment (left panel) and the logarithmic display used to calculate the K_d (right panel).

Left: Fluorescence anisotropy raw data

Right: Fluorescence anisotropy raw data with Log10 x axis for calculation

Data table

c [μM]	c [mM]	c [M]	c [Log10(M)]
0,4	0,0004	0,0000004	-6,39794
0,8	0,0008	0,0000008	-6,09691
1,2	0,0012	0,0000012	-5,9208188
1,6	0,0016	0,0000016	-5,79588
2	0,002	0,000002	-5,69897
2,5	0,0025	0,0000025	-5,60206
3	0,003	0,000003	-5,5228787
3,5	0,0035	0,0000035	-5,455932
4	0,004	0,000004	-5,39794
6	0,006	0,000006	-5,2218487
8	0,008	0,000008	-5,09691
10	0,01	0,00001	-5
15	0,015	0,000015	-4,8239087
20	0,02	0,00002	-4,69897
25	0,025	0,000025	-4,60206

Competition experiment of Figure 2A: Despite the new fluorescence polarization data, it remains clear that adding as much as 65 micromolar of Ame1[1-20] caused only a very minor drop in the amount of MN that was pulled down here by Mif2-N. Thus the competition between these two N-terminal peptides for binding MN is relatively weak by this measure. Why don't the two measures (pull-down versus fluorescence polarization) agree on this point? Given the inconsistency, it seems overstated to say that these interactions are "proved to be mutually exclusive" (page 5).

We view this as a limitation of the competition experiment coupled with solid phase binding to the GST-Fusions. Note that the free Ame1 peptide is also not very efficiently competing with GST-Ame1N for MN binding (lane 2 versus 3 in Figure 2A). The high local concentration of one binding partner on the beads probably prevents quantitative statements here. These problems are overcome in the FP assay, where all binding partners are in solution. To reflect the remaining concerns of this Referee we have changed the wording in the text from 'which we prove to be mutually exclusive' to 'which our data indicates to be mutually exclusive'.

Typo, page 7: "conserved Glycin" should be Glycine.

We corrected the typo.

About the Western blot in Figure 3D: "The inability of these alleles to support growth was not due to lack of protein expression, as shown by western blot analysis (Figure 3d)." This seems to be an over-interpretation. The blot shows that levels of all the mutants are clearly reduced compared to wildtype levels. Thus, contrary to what is claimed here in the text, it seems that the lower levels of all the mutant proteins could indeed explain their inability to support growth.

This is a fair point. We have changed the wording in the text to state that "The inability of these alleles to support growth was not due to lack of protein expression, **although the steady state level was reduced compared to the wild-type as shown by western blotting**". As a side note: We interpret the reduced steady state level to be a consequence of the inability to correctly localize to the kinetochore (The reason is that we have unpublished observations showing that non-kinetochore localized subunits are preferentially degraded by the cellular machinery). In addition, from other experiments in the lab we would assume that even the relatively low level of Mtw1-8A would exceed that of Dsn1 (a subunit from the same complex), making it unlikely that Mtw1 simply by its level would become limiting for Mtw1 complex assembly under these conditions.

On page 17, in the last three sentences before Discussion, it seems the references to Figures 8G and 8F are inadvertently swapped.

Thanks, we changed that.

Page 30, methods for fluorescence polarization are omitted: "...with a X and the software X". Also on subsequent page, what is meant by "preceded by excitation wavelength..."?

We have corrected these points.

Additional changes

In addition to these requested changes, we made the following minor corrections or additions to the manuscript:

1. Abstract: We slightly changed the wording of the abstract, stating that the described mechanism "contributes" to preventing unscheduled kinetochore assembly, and stating that we "propose" that auto-inhibition constitutes a key concept. This carefully softens the claim and addresses concerns of reviewer 3 regarding "bold" language.

2. Figure 5D: We quantified the abundance of crosslinks using "number of spectral counts" as an established semi-quantitative measure. This highlights the difference between crosslinks in free Mif2 versus Cse4n-bound Mif2. We have amended the results section accordingly.

Thank you for submitting your final revised manuscript for our consideration. I am pleased to inform you that we have now accepted it for publication in The EMBO Journal.

Corresponding Author Name: Stefan Westermann

Journal Submitted to: The Embo Journal

Manuscript Number: EMBOJ-2019-102938R